# ON LOW RANK DIRECTED ACYCLIC GRAPHS AND CAUSAL STRUCTURE LEARNING

## ABSTRACT

Despite several important advances in recent years, learning causal structures represented by directed acyclic graphs (DAGs) remains a challenging task in high dimensional settings when the graphs to be learned are not sparse. In this paper, we propose to exploit a low rank assumption regarding the (weighted) adjacency matrix of a DAG causal model to mitigate this problem. We demonstrate how to adapt existing methods for causal structure learning to take advantage of this assumption and establish several useful results relating interpretable graphical conditions to the low rank assumption. In particular, we show that the maximum rank is highly related to hubs, suggesting that scale-free networks which are frequently encountered in real applications tend to be low rank. We also provide empirical evidence for the utility of our low rank adaptations, especially on relatively large and dense graphs. Not only do they outperform existing algorithms when the low rank condition is satisfied, the performance is also competitive even though the rank of the underlying DAG may not be as low as is assumed.

## 1 INTRODUCTION

An important goal in many sciences is to discover the underlying causal structures in various domains, both for the purpose of explaining and understanding phenomena, and for the purpose of predicting effects of interventions (Pearl, 2009). Due to the relative abundance of passively observed data as opposed to experimental data, how to learn causal structures from purely observational data has been vigorously investigated (Peters et al., 2017; Spirtes et al., 2000). In this context, causal structures are usually represented by directed acyclic graphs (DAGs) over a set of random variables.

For this task, existing methods can be roughly categorized into two classes: constraint- and score-based. The former use statistical tests to extract from data a number of constraints in the form of conditional (in)dependence and seek to identify the class of causal structures compatible with those constraints (Meek, 1995; Spirtes et al., 2000; Zhang, 2008). The latter employ a score function to evaluate candidate causal structures relative to data and seek to locate the causal structure (or a class of causal structures) with the optimal score. Due to the combinatorial nature of the acyclicity constraint (Chickering, 1996; He et al., 2015), most score-based methods rely on local heuristics to perform the search. A particular example is the greedy equivalence search (GES) algorithm (Chickering, 2002) that can find an optimal solution with infinite data and proper model assumptions.

Recently, Zheng et al. (2018) introduced a smooth acyclicity constraint w.r.t. graph adjacency matrix, and the task on linear data models was then formulated as a continuous optimization problem with least-squares loss. This change of perspective allows using deep learning techniques to model causal mechanisms and has already given rise to several new algorithms for causal structure learning with non-linear data, e.g., Yu et al. (2019); Ng et al. (2019b;a); Ke et al. (2019); Lachapelle et al. (2020); Zheng et al. (2020), among others. While these new algorithms represent the current state of the art in many settings, their performance generally degrades when the target DAG becomes large and relatively dense, as seen from the empirical results reported in the referred works and also in this paper. This issue is of course a challenge to other approaches. Ramsey et al. (2017) proposed fast GES for impressively large problems, but it works reasonably well only when the large structure is very sparse. The max-min hill-climbing (MMHC) (Tsamardinos et al., 2006) relies on local learning methods that often do not perform well when the target node has a large neighborhood. How to improve the performance on relatively large and dense DAGs is therefore an important question.

In this work, we study the potential of exploiting a kind of *low rank* assumption on the DAG structure to help address this problem. The rank of a graph that concerns us is the algebraic rank of its associated weighted adjacency matrix. Similar to the role of a sparsity assumption on graph structures, we treat the low rank assumption as methodological and it is not restricted to a particular DAG learning method. However, unlike sparsity assumption, it is much less apparent when DAGs tend to be low rank and how low rank DAGs behave. Thus, besides demonstrating the utility of exploiting a low rank assumption in causal structure learning, another important goal is to improve our understanding of the low rank assumption by relating the rank of a graph to its graphical structure. Such a result also enables us to characterize the rank of a graph from several structural priors and helps to choose rank related hyperparameters for the learning algorithm. Our contributions are summarized as follows:

- We show how to adapt existing causal structure learning methods to take advantage of the low rank assumption, and provide a strategy to select rank related hyperparameters utilizing the lower and upper bounds on the true rank, if they are available.

- To improve our understanding of low rank DAGs, we establish some lower bounds on the rank of a DAG in terms of simple graphical conditions, which imply necessary conditions for DAGs to be low rank.

- We also show that the maximum possible rank of weighted adjacency matrices associated with a directed graph is highly related to hubs in the graph, which suggests that scale-free networks tend to be low rank. From this result, we derive several graphical conditions to bound the rank of a DAG from above, providing simple sufficient conditions for low rank.

- Empirically, we demonstrate that the low rank adaptations are indeed useful. Not only do they outperform the original algorithms when the low rank condition is satisfied, the performance is also very competitive even when the true rank is not as low as is assumed.

**Related Work**    The low rank assumption is frequently adopted in graph-based applications (Smith et al., 2012; Zhou et al., 2013; Yao & Kwok, 2016; Frot et al., 2019), matrix completion and factorization (Recht, 2011; Koltchinskii et al., 2011; Cao et al., 2015; Davenport & Romberg, 2016), network sciences (Hsieh et al., 2012; Huang et al., 2013; Zhang et al., 2017) and so on, but to our best knowledge, has not been used on the DAG structures in the context of learning causal DAGs. We notice two works Barik & Honorio (2019); Tichavský & Vomlel (2018) that assume low rank conditional probability tables in learning Bayesian networks, which are different from ours.

Also related are existing works that studied the rank of real weighted matrices described by a given simple directed/undirected graph. However, most works only considered the zero-nonzero pattern of *off-diagonal* entries (see, e.g., Fallat & Hogben (2007); Hogben (2010); Mitchell et al. (2010)), whereas we also take into account the diagonal entries. This difference is crucial: if one only considers the off-diagonal entries, then the maximum rank over all possible weighted matrices is trivial and is always equal to the number of vertices. Consequently, many works focus on the minimum rank of a given graph, but to characterize exactly the minimum rank remains open, except for some special graph structures like trees (Hogben, 2010). Apart from these works, Edmonds (1967) studied algebraically the maximum rank for matrices with a common zero-nonzero pattern. In Section 4, we use this result to relate the maximum possible rank to a more interpretable graphical condition, which further implies several structural conditions of DAGs that may be easier to obtain in practice.

## 2    PRELIMINARIES

### 2.1    GRAPH TERMINOLOGY

A graph $\mathcal{G}$ is defined as a pair $(\mathbf{V}, \mathbf{E})$, where $\mathbf{V} = \{X_1, X_2, \cdots, X_d\}$ is the vertex set and $\mathbf{E} \subset \mathbf{V}^2$ denotes the edge set. We are particularly interested in directed (acyclic) graphs in the context of causal structure learning. For any $\mathbf{S} \subset \mathbf{V}$, we use $\mathrm{pa}(\mathbf{S}, \mathcal{G})$, $\mathrm{ch}(\mathbf{S}, \mathcal{G})$, and $\mathrm{adj}(\mathbf{S}, \mathcal{G})$ to denote the union of all parents, children, and adjacent vertices of the nodes of $\mathbf{S}$ in $\mathcal{G}$, respectively. A graph is called weighted if every edge in the graph is associated with a non-zero value. We will work with weighted graphs and treat unweighted graphs as a special case where the edge weights are set to 1. Weighted graphs can be treated algebraically via weighted adjacency matrices. Specifically, the weighted adjacency matrix of a weighted graph $\mathcal{G}$ is a matrix $W \in \mathbb{R}^{d \times d}$, where $W(i, j)$ is the weight of edge $X_i \to X_j$ and $W(i, j) \neq 0$ if and only if $X_i \to X_j$ exists in $\mathcal{G}$. The binary adjacency

matrix $A \in \{0, 1\}^{d \times d}$ is such that $A(i, j) = 1$ if $X_i \to X_j$ in $\mathcal{G}$ and $A(i, j) = 0$ otherwise. The rank of a weighted graph is defined as the rank of the associated weighted adjacency matrix.

## 2.2 Causal Structure Learning and Recent Gradient-based Methods

A commonly used model in causal structure learning is the structural equation model (SEM) that describes data generating procedure. In a slight abuse of notation, we also use $X_i$'s to denote random variables associated with the nodes in a graph $\mathcal{G}$. Assuming $\mathcal{G}$ being a DAG, then the SEM is given by

$$X_i = f_i \left( \text{pa}(X_i, \mathcal{G}), \epsilon_i \right), \ \ i = 1, 2, \ldots, d,$$

where $f_i$ is a deterministic function and $\epsilon_i$'s are jointly independent noises. The SEM induces a marginal distribution $P(X)$ over $X = [X_1, X_2, \cdots, X_d]^T$, and $\mathcal{G}$ and $P(X)$ are said to form a causal Bayesian network (Pearl, 2009; Spirtes et al., 2000). The problem of causal structure learning is to infer the underlying causal DAG $\mathcal{G}$ based on the marginal distribution $P(X)$, or more practically, an empirical version consisting of a number of i.i.d. observations from $P(X)$.

We next briefly review recently developed gradient-based methods that rely on a smooth characterization of acyclicity of directed graphs. These methods aim to find a DAG that optimizes a score function and can be categorized into two classes. The first class of methods explicitly associates the target causal model with a weighted adjacency matrix $W$ and then estimate $W$ by solving optimization problems in the following form:

$$\min_{W, \phi} \mathbb{E}_{X \sim P(X)} \mathcal{S}\left( X, h(X; W, \phi) \right), \ \text{subject to trace} \left( e^{W \circ W} \right) - d = 0, \tag{1}$$

where $h : \mathbb{R}^d \to \mathbb{R}^d$ is a model function parameterized by $W$ (and other possible parameter $\phi$) that aims to reconstruct $X$, $\mathcal{S}(\cdot, \cdot)$ denotes a score function between the true and reconstructed variables, notation $\circ$ denotes the element-wise product, and $e^M$ is the matrix exponential of a square matrix $M$. The constraint was proposed by Zheng et al. (2018), which is smooth and holds if and only if $W$ indicates a DAG. Methods in this class include: NOTEARS (Zheng et al., 2018), which targets linear models, with $h(X; W, \phi) = W^T X$ and $\mathcal{S}(\cdot, \cdot)$ being the Frobenius norm or equivalently the least-squares loss; and DAG-GNN (Yu et al., 2019) and the graph autoencoder approach (Ng et al., 2019b), where neural networks are used for the function $h$ with $\phi$ being the weights of neural networks, and the score function can be chosen as the evidence lower bound (Kingma & Welling, 2013). A sparsity inducing term may be further added when the causal graph is assumed to be sparse. These objectives are equivalent to or are variants of some well studied score functions like the penalized maximum likelihood (Chickering, 2002; Van de Geer et al., 2013; Loh & Bühlmann, 2014).

The second class uses certain functions, with parameter $\theta$, to construct a weighted adjacency matrix $W(\theta)$ (or a binary one $A(\theta)$) to represent the causal structure. These methods can be summarized as

$$\min_{\theta, \phi} \mathbb{E}_{X \sim P(X)} \mathcal{S}\left( X, h(X; W(\theta), \phi) \right), \ \text{subject to trace} \left( e^{W(\theta) \circ W(\theta)} \right) - d = 0. \tag{2}$$

For example, GraN-DAG (Lachapelle et al., 2020) and NOTEARS-MLP (Zheng et al., 2020) respectively use neural network path products and partial derivatives between variables to construct $W(\theta)$. The binary matrix $A(\theta)$ can be obtained by sampling according to some distributions with learnable parameters, as used by Kalainathan et al. (2018); Ke et al. (2019); Ng et al. (2019a); Zhu et al. (2020).

Before ending this section, we remark that while the gradient-based methods intend to learn a causal DAG, the learned DAG may not be identical to the underlying one for general SEMs due to the Markov equivalence (Spirtes et al., 2000; Peters et al., 2017). For such cases, one may convert the obtained DAG to its corresponding Completed Partially Directed Acyclic Graph (CPDAG) as the estimate. Nevertheless, if the SEM is identifiable and a proper score function is used, then the exact solution to the optimization problem is consistent, i.e., same as the true graph with probability 1; see, e.g., Shimizu et al. (2006); Peters & Bühlmann (2013); Peters et al. (2014); Zhang & Hyvärinen (2009). For further details and other technical issues like parameter optimization of the gradient-based methods, we refer the reader to the cited works and references therein.

## 3 Exploiting Low Rank Assumption in Causal Structure Learning

This section shows how to adapt existing gradient-based methods to take advantage of the low rank assumption, by providing a way for each class to utilize this assumption using techniques from the

matrix completion literature. We remark that our adaptations with the low rank assumption are not restricted to a particular learning algorithm; other DAG learning methods may potentially combine one of the proposed modifications for learning low rank causal graphs, too.

**Matrix Factorization**   Since the weighted adjacency matrix $W$ is explicitly optimized in the first class of methods, we can then apply the matrix factorization technique. Specifically, with an estimate $\hat{r}$ for the graph rank, we can factorize $W$ as $W = UV^T$ with $U, V \in \mathbb{R}^{d \times \hat{r}}$. Problem (1) is then to optimize $U$ and $V$ that minimizes the score function under the DAG constraint, and has the same solution $W$ (obtained from the product $UV^T$) as the original one if $\hat{r}$ is greater than or equal to the true rank. Furthermore, if $\hat{r} \ll d$, we have a much reduced number of parameters to optimize.

**Nuclear Norm**   For the second class of methods, the adjacency matrix $W(\theta)$ is not an explicit parameter to be optimized. In such a case, we can adopt a commonly used technique to add a nuclear norm term $\lambda\|W(\theta)\|_*$, with $\lambda > 0$ being a tuning parameter, to the objective to induce low-rankness.

The optimization procedures in these recent structure learning methods can directly incorporate the two adaptations as they are all gradient-based, though some extra care needs to be taken. Appendix C provides a detailed description of the optimization procedure and our implementation. The second approach is also feasible for the first class of methods, but we find that it does not work as well as the matrix factorization approach, possibly due to the singular value decomposition to compute the (sub-)gradient w.r.t. $W$ at each optimization step.

An acute reader may have noticed that we assumed a proper rank estimate $\hat{r}$ or a proper penalty parameter $\lambda$. Yet knowing exactly the rank of the graph to be learned can be difficult in practice. Similar to the sparsity assumption, one may determine the hyperparameters $\hat{r}$ and $\lambda$ assisted by a validation dataset (or by cross-validation if the observed dataset is not sufficiently large). Alternatively, we can try different choices of the hyperparameters and then apply traditional score-based method where the search space is restricted to the resulting DAGs. However, since we are more concerned with relatively large and dense problems, the possible ranks may be too many to choose. As such, a lower bound $r_l$ and an upper bound $r_u$ on the graph rank would be beneficial—we need only consider ranks in $[r_l, r_u]$ in the matrix factorization method, while the bounds are still useful by providing qualitative information for the nuclear norm approach: the lower an upper bound, the higher the tuning parameter $\lambda$ should be chosen. Moreover, a lower bound can also justify the low rank assumption, i.e., if the lower bound is high, then the low rank assumption is likely to fail to hold.

## 4   GRAPHICAL BOUNDS ON RANKS

Obtaining exact algebraic information of a DAG such as its rank and eigenvalues may be infeasible in practice, because it may require a full knowledge of the graph to be learned. On the other hand, structural information, such as graph connectivity, distributions of in-degrees and out-degrees, and an estimate of number of hubs, is sometimes more accessible. As such, this section is devoted to relating the rank of a graph to more easily interpretable graphical conditions, for the sake of a better understanding of what kinds of DAGs tend to satisfy the low rank assumption and for lower and upper bounds on the graph rank from certain structural priors.

### 4.1   PROBLEM SETTING

Consider a DAG $\mathcal{G} = (\mathbf{V}, \mathbf{E})$ with weighted adjacency matrix $W$ and binary adjacency matrix $A$. We aim to seek upper and lower bounds on $\mathrm{rank}(W)$ using only the graphical structure. Specifically, we focus on the weighted adjacency matrices with the same binary adjacency matrix $A$, i.e., $\mathcal{W}_A = \{W \in \mathbb{R}^{d \times d}; \ \mathrm{sign}(|W|) = A\}$, where $\mathrm{sign}(\cdot)$ and $|\cdot|$ are point-wise sign and absolute value functions, respectively. Notice that there exist trivial upper bound $d - 1$ and lower bound $0$ for any DAG, but they are generally too loose for our purpose. In the following, we investigate the maximum rank $\max\{\mathrm{rank}(W); W \in \mathcal{W}_A\}$ and minimum rank $\min\{\mathrm{rank}(W); W \in \mathcal{W}_A\}$ to find tighter upper and lower bounds for any $W \in \mathcal{W}_A$. Before introducing two useful graph concepts, we comment that low rank DAGs are not necessarily sparse and vice versa; see a discussion in Appendix A.

**Definition 1** (Height). *Given a DAG $\mathcal{G} = (\mathbf{V}, \mathbf{E})$ and a vertex $X_i \in \mathbf{V}$, the height of $X_i$, denoted by $l(X_i)$, is defined as the length of the longest directed path starting from $X_i$. The height of $\mathcal{G}$, denoted by $l(\mathcal{G})$, is the length of the longest path in $\mathcal{G}$.*

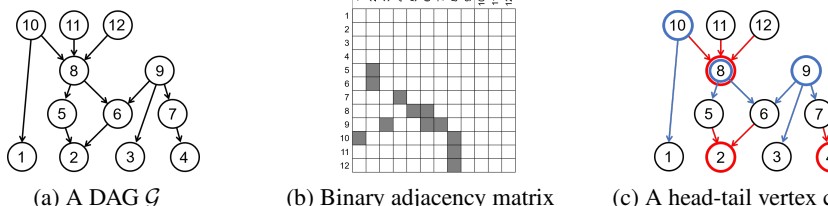

(a) A DAG $\mathcal{G}$       (b) Binary adjacency matrix       (c) A head-tail vertex cover

Figure 1: A DAG $\mathcal{G}$ with 12 vertices, 12 edges and height 3, where $\mathbf{V}_0 = \{X_1, X_2, X_3, X_4\}$, $\mathbf{V}_1 = \{X_5, X_6, X_7\}$, $\mathbf{V}_2 = \{X_8, X_9\}$, and $\mathbf{V}_3 = \{X_{10}, X_{11}, X_{12}\}$.

**Definition 2** (Head-tail vertex cover). *Let $\mathcal{G} = (\mathbf{V}, \mathbf{E})$ be a directed graph and $\mathbf{H}, \mathbf{T}$ be two subsets of $\mathbf{V}$. $(\mathbf{H}, \mathbf{T})$ is called a head-tail vertex cover of $\mathcal{G}$ if every edge in $\mathcal{G}$ has its head vertex in $\mathbf{H}$ or its tail vertex in $\mathbf{T}$. The size of a head-tail vertex cover $(\mathbf{H}, \mathbf{T})$ is defined as $|\mathbf{H}| + |\mathbf{T}|$.*

As an example, Figure 1c is a head-tail vertex cover of $\mathcal{G}$ in Figure 1a, where $\mathbf{H} = \{X_2, X_4, X_8\}$ (red nodes) and $\mathbf{T} = \{X_8, X_9, X_{10}\}$ (blue nodes). The size of this vertex cover is 6.

## 4.2 LOWER BOUNDS

We first study lower bounds on the rank of a weighted DAG. Define $\mathbf{V}_{-1} = \varnothing$ and $\mathbf{V}_s = \{X_i; l(X_i) = s\}$ for $s = 0, 1, \ldots, l(\mathcal{G})$. Denote by $\mathcal{G}_{s,s-1}$ the induced subgraph of $\mathcal{G}$ over $\mathbf{V}_s \cup \mathbf{V}_{s-1}$. Let $\mathcal{C}(\mathcal{G}_{s,s-1})$ be the set of non-singleton connected components of $\mathcal{G}_{s,s-1}$ and $|\mathcal{C}(\mathcal{G}_{s,s-1})|$ the cardinality. We have the following lower bounds.

**Theorem 1.** *Let $\mathcal{G}$ be a DAG with binary adjacency matrix $A$. Then*

$$\min\{\mathrm{rank}(W)\,;\ W \in \mathcal{W}_A\} \geq \sum_{s=1}^{l(\mathcal{G})} |\mathcal{C}(\mathcal{G}_{s,s-1})| \geq l(\mathcal{G}). \tag{3}$$

All the proofs in this paper are provided in Appendix B. Theorem 1 shows that $\mathrm{rank}(W)$ is greater than or equal to the sum of the number of non-singleton connected components in each $\mathcal{G}_{s,s-1}$. As $\mathcal{G}_{s,s-1}$ has at least one non-singleton connected component, we obtain the second inequality. In other words, the rank of a weighted DAG is at least as high as the length of the longest directed path. As an example, consider the graph shown in Figure 1. One can verify that $\min\{\mathrm{rank}(W); W \in \mathcal{W}_A\} = 6$, $|\mathcal{C}(\mathcal{G}_{1,0})| = 2$, $|\mathcal{C}(\mathcal{G}_{2,1})| = 1$, $|\mathcal{C}(\mathcal{G}_{3,2})| = 1$, and $l(\mathcal{G}) = 3$. Thus, we have $\min\{\mathrm{rank}(W); W \in \mathcal{W}_A\} = 6 > 2 + 1 + 1 = 4 > 3$. We remark that the bounds in Theorem 1 may be loose in some cases. To characterize the minimum rank exactly is an on-going research problem (Hogben, 2010).

## 4.3 UPPER BOUNDS

We turn to the more important issue for our purpose, regarding upper bounds on $\mathrm{rank}(W)$. The next theorem shows that $\max\{\mathrm{rank}(W); W \in \mathcal{W}_A\}$ can be characterized exactly in graphical terms.

**Theorem 2.** *Let $\mathcal{G}$ be a directed graph with binary adjacency matrix $A$. Then $\max\{\mathrm{rank}(W); W \in \mathcal{W}_A\}$ is equal to the minimum size of the head-tail vertex cover of $\mathcal{G}$, that is,*

$$\max\{\mathrm{rank}(W)\,;\ W \in \mathcal{W}_A\} = \min\{|\mathbf{H}| + |\mathbf{T}|\,;\ (\mathbf{H}, \mathbf{T}) \text{ is a head-tail vertex cover of } \mathcal{G}\}.$$

We comment that Theorem 2 holds for all directed graphs (not only DAGs), which may be of independent interest to other applications. A head-tail vertex cover of minimum size is called a minimum head-tail vertex cover, which in general is not unique. For a head-tail vertex cover $(\mathbf{H}, \mathbf{T})$, the vertices in $\mathbf{H}$ cover all the edges pointing towards these vertices while the vertices in $\mathbf{T}$ cover the edges pointing away. A head-tail cover of a relatively small size then indicates the presence of hubs, that is, vertices with relatively high in-degrees or out-degrees. Therefore, Theorem 2 suggests that the maximum rank of a weighted DAG is highly related to the presence of hubs: a DAG with many hubs tends to have low rank. Intuitively, a hub of high in-degree (out-degree) is a common effect (cause) of a number of direct causes (effect variables), comprising many V-structures (inverted V-structures). For example, in Figure 1a, $X_8$ is a hub of V-structures and $X_9$ is a hub of inverted V-structures.

Such features are fairly common in real graph structures. Appendix A presents a real network, called *pathfinder*, which describes the causal relations among 109 variables (Heckerman et al., 1992) with the center node being the parent of a large number of other nodes. The famous scale-free (SF) graphs also tend to have hubs. A scale-free graph is one whose distribution of degree $k$ follows a power law: $P(k) \sim k^{-\gamma}$, where $\gamma$ is the power parameter typically within $[2, 3]$ and $P(k)$ denotes the fraction of nodes with degree $k$ (Nikolova & Aluru, 2012). It is observed that many real-world networks are scale-free, and some of them, such as gene regulatory networks, protein networks, and financial system network, may be viewed as causal networks (Guelzim et al., 2002; Barabasi & Oltvai, 2004; Hartemink, 2005; Eguíluz et al., 2005; Gao & en Ren, 2013; Ramsey et al., 2017). In particular, Barabasi & Oltvai (2004) claimed that most protein networks, some of which are directed and acyclic due to irreversible reactions, are the results of growth processes and preferential attachments, probably due to the gene duplication.

Empirically, the ranks of scale-free graphs are relatively low, especially in comparison to Erdös-Rényi (ER) random graphs (Mihail & Papadimitriou, 2002). Figure 2 provides a simulated example where $\gamma$ is chosen from $\{2, 3\}$ and each reported value is over 100 random runs. As graph becomes denser, the graph rank also increases. However, for scale-free graphs with a relatively large $\gamma$, the increase of their ranks is much slower than that of Erdös-Rényi graphs; indeed, their ranks tend to stay fairly low even when the graph degree is large.

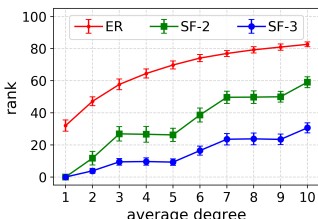

Figure 2: 100-node graphs.

Theorem 2 can also be used to generate a low rank graph, or more precisely, a random DAG with a given rank $r$ and a properly specified graph degree. Here we briefly describe the idea and leave the detailed algorithm to Appendix C.1: first generate a graph with $r$ edges and rank $r$; a random edge is sampled without replacement and would be added to the graph, if adding this edge does not increase the size of the minimum head-tail vertex cover; repeat the previous step until the pre-specified degree is reached or no edge could be added to the graph; finally, assign the edge weights randomly according to a continuous distribution and the weighted graph will have rank $r$ with high probability.

The next two theorems report some looser but simpler upper bounds on $\mathrm{rank}(W)$.

**Theorem 3.** *Let $\mathcal{G}$ be a DAG with binary adjacency matrix $A$, and denote the set of vertices with at least one parent by $\mathbf{V}_{\mathrm{ch}}$ and those with at least one child by $\mathbf{V}_{\mathrm{pa}}$. Then we have*

$$\max\{\mathrm{rank}(W)\,;\, W \in \mathcal{W}_A\} \leq \begin{cases} \sum_{s=1}^{l(\mathcal{G})} \min\left(|\mathbf{V}_s|, |\mathrm{ch}(\mathbf{V}_s)|\right) \leq |\mathbf{V}_{\mathrm{pa}}|, \\ \sum_{s=0}^{l(\mathcal{G})-1} \min\left(|\mathbf{V}_s|, |\mathrm{pa}(\mathbf{V}_s)|\right) \leq |\mathbf{V}_{\mathrm{ch}}|, \\ |\mathbf{V}| - \max\{|\mathbf{V}_s|\,;\, 0 \leq s \leq l(\mathcal{G})\}. \end{cases} \quad (4)$$

Since $\mathbf{V}_{\mathrm{ch}}$ and $\mathbf{V}_{\mathrm{pa}}$ are the non-root and the non-leaf vertices, respectively, the first two inequalities of (4) indicate that the maximum rank is bounded from above by the number of non-root vertices and also by the number of non-leaf vertices. The last inequality of (4) is a generalization of the first two, which implies that the rank is likely to be low if most vertices have the same height.

**Theorem 4.** *Let $\mathcal{G}$ be a DAG with binary adjacency matrix $A$. Denote by $\mathrm{skeleton}(A)$ and $\mathrm{moral}(A)$ the binary adjacency matrices of the skeleton and moral graph of $\mathcal{G}$, respectively. Then we have*

$$\max\{\mathrm{rank}(W)\,;\, W \in \mathcal{W}_A\} \leq \max\{\mathrm{rank}(W)\,;\, \mathrm{sign}(|W|) = \mathrm{skeleton}(A)\}$$
$$\leq \max\{\mathrm{rank}(W)\,;\, \mathrm{sign}(|W|) = \mathrm{moral}(A)\}.$$

The skeleton of a DAG is the undirected graph obtained by removing all the arrowheads, and the moral graph is the undirected graph where two vertices are adjacent if they are adjacent or if they share a common child in the DAG. This result is useful when the skeleton or the moral graph can be accurately estimated and the corresponding rank is low. In practice, we may use all available structural priors to obtain upper bounds on the underlying rank and choose the lowest one as our estimate.

## 5 EXPERIMENTS

This section reports empirical results of the low rank adaptations of existing methods, compared with their original versions. We choose NOTEARS (Zheng et al., 2018) for linear SEMs by adopting the matrix factorization approach, denoted as NOTEARS-low-rank, and use the nuclear norm approach in combination with GraN-DAG (Lachapelle et al., 2020) for a non-linear data model. Again we remark that the two methods are only demonstrations of the utility of low rank assumption, which can be potentially combined with other methods as well. For more information, we also include several benchmark methods: fast GES (Ramsey et al., 2017), PC (Spirtes et al., 2000), MMHC (Tsamardinos et al., 2006), ICA-LiNGAM (Shimizu et al., 2006) specifically designed with non-Gaussian noises, for linear SEMs;[1] and DAG-GNN (Yu et al., 2019), NOTEARS-MLP (Zheng et al., 2020), and CAM (Bühlmann et al., 2014) for the non-linear case. Their implementations are described in Appendix C.

We consider randomly sampled DAGs with specified ranks (the generating procedure was described in Section 4.3 and is given as Algorithm 1 in Appendix C.1), scale-free graphs, and a real network structure. For linear SEMs, the weights are uniformly sampled from $[-2, -0.5] \cup [0.5, 2]$ and the noises are either standard Gaussian or standard exponential. For non-linear SEMs, we use additive Gaussian noise model with functions sampled from Gaussian processes with RBF kernel of bandwidth one. These data models are known to be identifiable (Shimizu et al., 2006; Peters & Bühlmann, 2013; Peters et al., 2014). From each SEM, we then generate $n = 3,000$ observations. We repeat ten times over different seeds for each experiment setting. Detailed information about the setup can be found in Appendix C.3. Below we mainly report structural Hamming distance (SHD) which takes into account both false positives and false negatives, and a smaller SHD indicates a better estimate.

### 5.1 LINEAR SEMS WITH RANK-SPECIFIED GRAPHS

We first consider linear SEMs on rank-specified graphs, with number of nodes $d \in \{100, 300\}$, rank $r = \lceil 0.1d \rceil$, and average degree $k \in \{2, 4, 6, 8\}$. The true rank is assumed to be known and is used as the rank parameter $\hat{r}$ in NOTEARS-low-rank. For a better visualization, Figure 3 only reports the average SHDs, while the true positive rate, false discovery rate, and running time are left to Appendix D. We also show the results after using the interquartile range rule to remove outlier SHDs. We observe that the low rank assumption can greatly improve the performance of NOTEARS, reducing the SHDs by at least a half. For this data model, the fast GES has much higher SHDs (see also Appendix D). PC is too slow (for example, it did not finish in 16 hours for a dataset with 100 nodes and degree 6), because some nodes may have a high in-degree. For the same reason, the skeleton may not be well estimated by MMHC; its performance is slightly worse than the fast GES and is not reported.

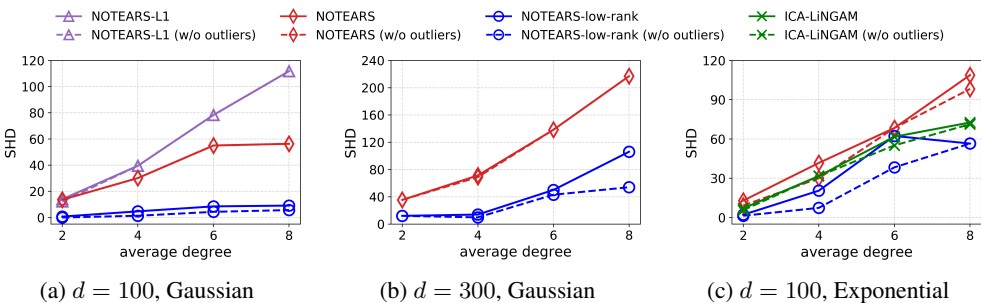

(a) $d = 100$, Gaussian       (b) $d = 300$, Gaussian       (c) $d = 100$, Exponential

Figure 3: Average SHDs on rank-specified graphs. The models are linear SEMs with (a)-(b) Gaussian noises, and (c) exponential noises. The true rank is assumed to be known.

For more information regarding the role of sparsity, we include NOTEARS with an $\ell_1$ penalty, named NOTEARS-L1. Here the $\ell_1$ penalty weight is chosen from $\{0.01, 0.02, 0.05, 0.1, 0.2, 0.5\}$. Instead

---

[1]Here we choose ICA-LiNGAM, other than alternative LiNGAM methods like DirectLiNGAM (Shimizu et al., 2011), based on our empirical observation. Specifically, an implementation of ICA-LiNGAM has a noticeably better performance than DirectLiNGAM for relatively dense graphs. Please find a detailed discussion and an empirical comparison in Appendix D.4.

of relying on an additional validation dataset, we treat NOTEARS-L1 favorably by picking the lowest SHD obtained from different weights for each dataset. As seen from Figure 3a, NOTEARS-L1 is slightly better than NOTEARS when the average degree is 2, but is largely outperformed with relatively dense graphs. This observation was also reported in Zheng et al. (2018). We conjecture that it is because our experiments consider relatively sufficient data and dense graphs. Moreover, the thresholding procedure controls false discoveries and may have a similar effect to the $\ell_1$ penalty.

Appendix D.1 studies graphs with higher ranks, where it is observed that the advantage of NOTEARS-low-rank over NOTEARS decreases when the rank of the underlying DAG increases. Nevertheless, NOTEARS-low-rank is still competitive when the true rank is $\lceil d/2 \rceil$ and the factorized matrix has the same number of parameters as NOTEARS. We also conduct an empirical analysis with different sample sizes in Appendix D.2, which shows that NOTEARS-low-rank performs reasonably well when the sample size is small and tends to have a better performance with a larger number of samples. Due to space limit, please find further details in the appendix.

## 5.2 LINEAR SEMs WITH SCALE-FREE GRAPHS

We next consider scale-free graphs with $d = 100$ nodes, average degree $k = 6$, and power $\gamma = 2.5$. For this experiment, the minimum, maximum, and mean ranks of generated graphs are $14$, $24$, and $18.7$, respectively. Here we choose the rank parameter $\hat{r}$ from $\{20, 30, 40\}$ for NOTEARS-low-rank. As seen from Figure 4, NOTEARS-low-rank with rank parameter $\hat{r} = 20$ performs the best, even though there are graphs with ranks greater than $20$.

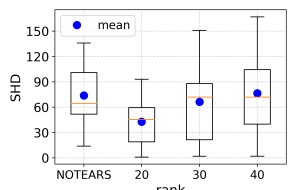

Figure 4: Scale-free graphs.

## 5.3 SENSITIVITY OF RANK PARAMETERS AND VALIDATION

So far we have assumed that the true rank or an accurate estimate is known. In this experiment, we conduct an empirical analysis with different rank parameters for linear Gaussian data model on rank-specified graphs with $100$ nodes, degree $8$, and rank $10$. We also include the validation based approach where $2,000$ samples are chosen as training dataset and the rest as validation dataset. We use the derived lower and upper bounds in Theorems 1 and 3 to obtain a range of possible rank parameters, assuming that the corresponding structural priors are available. Within this range, we then select 7 evenly distributed rank parameters used with NOTEARS-low-rank to learn causal graphs. Finally, we evaluate each learned DAG using the validation dataset and choose the DAG with the best score as our estimate.

As seen from Figure 5, NOTEARS-low-rank performs the best when the rank parameter is identical to the true rank, while the rank parameter chosen by validation has almost the same performance. Compared with NOTEARS on the same datasets, the low rank version performs well across a range of rank parameters. Although this validation approach increases the total running time that depends on the number of candidate rank parameters, we believe that it is acceptable given the gained accuracy and also the fact that this strategy has been frequently adopted for tuning hyperparameters in practice.

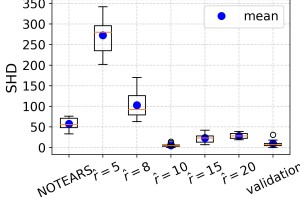

Figure 5: Different rank parameters.

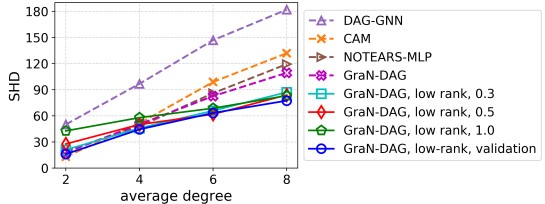

Figure 6: Non-linear SEMs.

## 5.4 NON-LINEAR SEMs

For non-linear data models, we pick rank-specified graphs with 50 nodes, rank 5, and average degree $k \in \{2, 4, 6, 8\}$. To our knowledge, the selected benchmark methods CAM, NOTEARS-MLP, and

GraN-DAG are state-of-the-art methods on this data model. As a demonstration of the low rank assumption, we apply the nuclear norm approach to GraN-DAG and choose from $\{0.3, 0.5, 1.0\}$ as penalty weights. For validation, we use the same splitting ratio as in Section 5.3 and consider more penalty weights from $\{0.1, 0.2, 0.3, 0.5, 1, 2, 5\}$. Similarly, the learned graph that achieves the best score on the validation dataset is chosen as final estimate. Figure 6 (and Appendix D.6 with a more detailed result) shows that adding a nuclear norm can improve the performance of GraN-DAG across a large range of weights when the graph is relatively dense. For degree $8$, the low rank version with validation achieves average SHD $77.4$, while the SHDs of CAM, NOTEARS-MLP, and original GraN-DAG are $131.9$, $119.4$, and $109.4$, respectively.

### 5.5 REAL NETWORK

We apply the proposed method to the *arth150* gene network, which is a DAG containing $107$ genes and $150$ edges. Its maximum rank is $40$. Since the real dataset has only $22$ samples, we instead use simulated data from linear Gaussian SEMs. We pick $\hat{r}$ from $\{36, 40, 44\}$ and also use validation to select the rank parameter. We apply NOTEARS-L1 where the $\ell_1$ penalty weight is chosen from $\{0.05, 0.1, 0.2\}$, and similarly treat this method favorably by picking the lowest SHD for each dataset. The mean and median SHDs are shown in Figure 7. Using Student's t-test, we find that with significance level $0.1$, the results

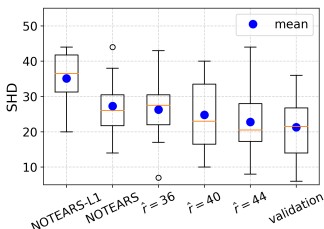

Figure 7: Real network.

obtained with $\hat{r} = 44$ and the validation approach are significantly better than NOTEARS. This experiment demonstrates again the utility of the low rank assumption, even when the true rank of the graph is not very low.

## 6 CONCLUDING REMARKS

This paper studies the potential of low rank assumption in causal structure learning. Empirically, we show that the low rank adaptations perform noticeably better than existing algorithms when the low rank condition is satisfied, and also deliver competitive performances when the rank is not as low as is assumed. Theoretically, we provide an improved understanding of what kinds of graphs tend to be low rank and a possibility to obtain bounds on the underlying rank from several structural priors.

We treat the present work as our first step to incorporate low-rankness into causal DAG learning. A future direction is to approximate a high rank DAG with a low rank one (possibly adding an additional DAG that is sparse). While there is a rich literature on low rank approximations of matrices and combining low-rankness with sparsity, it is non-trivial to us to conclude under what conditions such an approximation is guaranteed to be effective to learn causal DAGs. Another direction is to compare the low rank assumption to other structural or parametric priors affecting model selection through marginal likelihood (Eggeling et al., 2019; Silander et al., 2007). Finally, it is also interesting to investigate if a low rank DAG model implies any useful behavior in the data.

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

# Appendix

## A EXAMPLES AND DISCUSSIONS

We provide more examples and discussions in this section.

**Minimum rank of the graph in Figure 1**  We first show that the minimum rank of the DAG structure in Figure 1 is 6. It is clear that the 6-th to 10-th rows of $A$ are always linearly independent, so it suffices to show that the 11-th row is linearly independent of the 6-th to 10-th rows. To see this, notice that if the 11-th row is a linear combination of the 6-th to 10-th rows, then $A(11, 1)$ would be non-zero, which is a contradiction.

**The *pathfinder* and *arth150* networks**  Figure 8 visualizes the *pathfinder* and *arth150* networks that are mentioned in Sections 4.3 and 5, respectively. Both networks can be found at `http://www.bnlearn.com/bnrepository`. As one can see, these two networks contain hubs: the center note in the *pathfinder* network has a large number of children, while the *arth150* network contains many 'small' hubs, each of which has $5 \sim 10$ children. We also notice that nearly all the hubs in the two networks have high out-degrees.

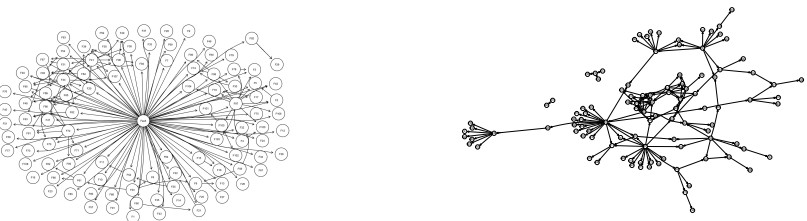

Figure 8: The *pathfinder* (left) and *arth150* (right) networks.

**Sparse DAGs and low rank DAGs**  A sparse DAG does not necessarily indicate a low rank DAG, and vice versa. For example, a directed linear graph with $d$ vertices has only $d - 1$ edges, i.e. $X_1 \to X_2 \to \cdots \to X_d$, while the rank of its binary adjacency matrix is $d - 1$. According to Theorems 1 and 2, the maximum and minimum ranks of a directed linear graph are equal to its number of edges. Thus, directed linear graphs are sparse but have high ranks. On the other hand, for some non-sparse graphs, we can assign the edge weights so that the resulting graphs have low ranks. A simple example would be a fully connected directed balanced bipartite graph, as shown in Figure 9. The definition of bipartite graphs can be found in Appendix B.1. A bipartite graph is called balanced if its two parts contain the same number of vertices. The rank of a fully connected balanced bipartite graph with $d$ vertices is 1 if all the edge weights are the same (e.g., the binary adjacency matrix), but the number of edges is $d^2/4$.

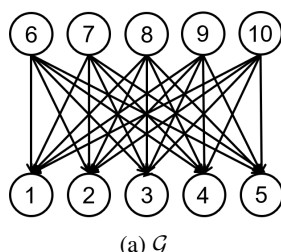         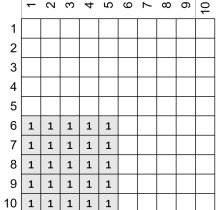

(a) $\mathcal{G}$                    (b) Binary adjacency matrix

Figure 9: A fully connected directed balanced bipartite graph $\mathcal{G}$ and its binary adjacency matrix.

We also notice that there exist some connections between the maximum rank and the graph degree, or more precisely, the total number of edges in the graph, according to Theorem 2. Intuitively, if the

graph is dense, then we need more vertices to cover all the edges. Thus, the size of the minimum head-tail vertex cover should be large. Explicitly providing a formula to characterize these two graph parameters is an interesting problem, which will be explored in the future.

## B  PROOFS

In this section, we present proofs for the theorems given in the main content.

### B.1  PRELIMINARIES

A bipartite graph is a graph whose vertex set $\mathbf{V}$ can be partitioned into two disjoint subsets $\mathbf{V}_0$ and $\mathbf{V}_1$, such that the vertices within each subset are not adjacent to one another. $\mathbf{V}_0$ and $\mathbf{V}_1$ are called the parts of the graph. A matching of a graph is a subset of its edges where no two of them share a common endpoint. A vertex cover of a graph is a subset of the vertex set where every edge in the graph has at least one endpoint in the subset. The size of a matching (vertex cover) is the number of edges (vertices) in the matching (vertex cover). A maximum matching of a graph is a matching of the largest possible size and a minimum vertex cover is a vertex cover of the smallest possible size. An important result about bipartite graphs is König's theorem (Dénes, 1931), which states that the size of a minimum vertex cover is equal to the size of a maximum matching in a bipartite graph.

Based on the heights of vertices in $\mathbf{V}$, we can define a weak ordering among the vertices: $X_i \succ X_j$ if and only if $l(X_i) > l(X_j)$, and $X_i \sim X_j$ if and only if $l(X_i) = l(X_j)$. Given this weak ordering, we can group the vertices by their heights, and the resulting graph shows a hierarchical structure; see Figure 1 in the main text for an example. This hierarchical representation has some simple and nice properties. Let $\mathbf{V}_s = \{X_i; l(X_i) = s\}$, $s = 0, 1, \ldots, l(\mathcal{G})$, and let $\mathbf{V}_{-1} = \varnothing$. We have: (1) for any given $s \in \{0, 1, \ldots, l(\mathcal{G})\}$ and two distinct vertices $X_1, X_2 \in \mathbf{V}_s$, $X_1$ and $X_2$ are not adjacent, and (2) for any given $s \in \{1, 2, \ldots, l(\mathcal{G})\}$ and $X_i \in \mathbf{V}_s$, there is at least one vertex in $\mathbf{V}_{s-1}$ which is a child of $X_i$. If we denote the induced subgraph of $\mathcal{G}$ over $\mathbf{V}_s \cup \mathbf{V}_{s-1}$ by $\mathcal{G}_{s,s-1}$, then $\mathcal{G}_{s,s-1}$ is a bipartite graph with $\mathbf{V}_s$ and $\mathbf{V}_{s-1}$ as parts, and singletons in $\mathcal{G}_{s,s-1}$ (i.e., vertices that are not endpoints of any edge) only appear in $\mathbf{V}_{s-1}$.

For ease of presentation, we occasionally use index $i$ to represent variable $X_i$ in the following sections.

### B.2  PROOF OF THEOREM 1

*Proof.* Let $\mathcal{G} = (\mathbf{V}, \mathbf{E})$. Consider an equivalence relation, denoted by $\sim$, among vertices in $\mathbf{V}$ defined as follows: for any $X_i, X_j \in \mathbf{V}$, $X_i \sim X_j$ if and only if $l(X_i) = l(X_j)$ and $X_i$ and $X_j$ are connected. Here, connected means that there is a path between $X_i$ and $X_j$. Below we use $C(X_i)$ to denote the equivalence class containing $X_i$. Next, we define a weak ordering $\pi$ on $\mathbf{V}/\sim$, i.e., the equivalence classes induced by $\sim$, by letting $C(X_i) \succeq_\pi C(X_j)$ if and only if $l(X_i) \geq l(X_j)$. Then, we extend $\succeq_\pi$ to a total ordering $\rho$ on $\mathbf{V}/\sim$. The ordering $\rho$ also induces a weak ordering (denoted by $\bar{\rho}$) on $\mathbf{V}$: $X_i \succeq_{\bar{\rho}} X_j$ if and only if $C(X_i) \succ_\rho C(X_j)$. Finally, we extend $\bar{\rho}$ to a total ordering $\gamma$ on $\mathbf{V}$. It can be verified that $\gamma$ is a topological ordering of $\mathcal{G}$, that is, if we relabel the vertices according to $\gamma$, then $X_i \in \mathrm{pa}(X_j, \mathcal{G})$ if and only if $i > j$ and $X_i$ and $X_j$ are adjacent, and the adjacency matrix of $\mathcal{G}$ becomes lower triangular.

Assume that the vertices of $\mathcal{G}$ are relabeled according to $\gamma$ and we will consider the binary adjacency matrix $A$ of the resulting graph throughout the rest of this proof. Note that relabelling is equivalent to applying a permutation onto the adjacency matrix, which does not change the rank. Let $\mathbf{V}_0 = \{1, 2, \ldots, k_1 - 1\}$ for some $k_1 \geq 2$. Then the $k_1$-th row of $A$, denoted by $A(k_1, \cdot)$, is the first non-zero row vector of $A$. Letting $S = \{A(k_1, \cdot)\}$, then $S$ contains a subset of linearly independent vector(s) of the first $k_1$ rows of $A$. Suppose that we have visited the first $m$ rows of $A$ and $S = \{A(k_1, \cdot), A(k_2, \cdot), \ldots, A(k_t, \cdot)\}$ contains a subset of linearly independent vector(s) of the first $m$ rows of $A$, where $k_1 \leq m < d$. If $X_{m+1} \nsim X_{k_t}$, then we add $A(m+1, \cdot)$ to $S$; otherwise, we keep $S$ unchanged. We claim that the vectors in $S$ are still linearly independent after the above step. Clearly, if we do not add any new vector, then $S$ contains only linearly independent vectors. To show the other case, note that if $l(X_{m+1}) > l(X_{k_t}) \geq \cdots \geq l(X_{k_1})$, then there is an index $i \in \mathbf{V}_{l(X_{m+1})-1}$ such that $A(m+1, i) \neq 0$, by the definition of height. Since $l(X_{m+1}) > l(X_{k_t})$, we have

$l(X_{k_t}) \leq l(X_{m+1}) - 1$ and thus $A(k_j, i) = 0$ for all $j = 1, 2, \ldots, t$. Therefore, $A(m+1, \cdot)$ cannot be linearly represented by $\{A(k_j, \cdot); j = 1, 2, \ldots, t\}$ and the vectors in $S$ are linearly independent. On the other hand, if $l(X_{m+1}) = l(X_{k_t})$, then the definition of the equivalence relation $\sim$ implies that $X_{m+1}$ and $X_{k_t}$ are disconnected, which means that $X_{m+1}$ and $X_{k_t}$ do not share a common child in $\mathbf{V}_{l(X_{m+1})-1}$. Consequently, there is an index $i \in \mathbf{V}_{l(X_{m+1})-1}$ such that $A(m+1, i) \neq 0$ but $A(k_t, i) = 0$. Similarly, we can show that $A(k_j, i) = 0$ for all $j = 1, 2, \ldots, t$. Thus, the vectors in $S$ are still linearly independent.

After visiting all the rows in $A$, the number of vectors in $S$ is equal to $\sum_{s=1}^{l(\mathcal{G})} |\mathcal{C}(\mathcal{G}_{s,s-1})|$ based on the definition of $\sim$. The second inequality can be shown by noting that $\mathcal{C}(\mathcal{G}_{s,s-1})$ has at least one elements. The proof is complete. $\qquad\square$

### B.3 PROOF OF THEOREM 2

*Proof.* Denote the directed graph by $\mathcal{G} = (\mathbf{V}, \mathbf{E})$. Edmonds (1967, Theorem 1) showed that $\max\{\text{rank}(W); W \in \mathcal{W}_A\}$ is equal to the maximum number of nonzero entries of $A$, no two of which lie in a common row or column. Therefore, it suffices to show that the latter quantity is equal to the size of the minimum head-tail vertex cover. Let $\mathbf{V}' = \mathbf{V}'_0 \cup \mathbf{V}'_1$, where $\mathbf{V}'_0 = \mathbf{V} \times \{0\} = \{(X_i, 0); X_i \in \mathbf{V}\}$ and $\mathbf{V}'_1 = \mathbf{V} \times \{1\} = \{(X_i, 1); X_i \in \mathbf{V}\}$. Now define a bipartite graph $\mathcal{B} = (\mathbf{V}', \mathbf{E}')$ where $\mathbf{E}' = \{(X_i, 0) \to (X_j, 1); (X_i, X_j) \in \mathbf{E}\}$. Denote by $\mathcal{M}$ a set of nonzero entries of $A$ so that no two entries lie in the same row or column. Notice that $\mathcal{M}$ can be viewed as an edge set and no two edges in $\mathcal{M}$ share a common endpoint. Thus, $\mathcal{M}$ is a matching of $\mathcal{B}$. Conversely, it can be shown by similar arguments that any matching of $\mathcal{B}$ corresponds to a set of nonzero entries of $A$, no two of which lie in a common row or column. Therefore, $\max\{\text{rank}(W), W \in \mathcal{W}_A\}$ equals the size of the maximum matching of $\mathcal{B}$, and further the size of the minimum vertex cover of $\mathcal{B}$ according to König's theorem. Note that any vertex cover of $\mathcal{B}$ can be equivalently transformed to a head-tail vertex cover of $\mathcal{G}$, by letting $\mathbf{H}$ and $\mathbf{T}$ be the subsets of the vertex cover containing all variables in $\mathbf{V}'_0$ and of the vertex cover containing all variables in $\mathbf{V}'_1$, respectively. Thus, $\max\{\text{rank}(W), W \in \mathcal{W}_A\}$ is equal to the size of the minimum head-tail vertex cover. $\qquad\square$

### B.4 PROOF OF THEOREM 3

*Proof.* We start with the first inequality in Equation (4). Let $h_1, \ldots, h_p$ denote the heights where $|\mathbf{V}_s| < |\text{ch}(\mathbf{V}_s)|$, and $t_1, \ldots, t_q$ the height where $|\mathbf{V}_s| > |\text{ch}(\mathbf{V}_s)|$. Let $\mathbf{H} = \cup_{i=1}^p \mathbf{V}_{h_i}$ and $\mathbf{T} = \cup_{i=1}^q \mathbf{V}_{t_i}$. It is straightforward to see that $(\mathbf{H}, \mathbf{T})$ is a head-tail vertex cover. Thus, Equation (4) holds according to Theorem 2. The second inequality can be shown similarly and its proof is omitted. For the third inequality, let $m = \text{argmax}\{|\mathbf{V}_s| : 0 \leq s \leq l(\mathcal{G})\}$, and define $\mathbf{H} = \cup_{i>m} \mathbf{V}_i$ and $\mathbf{T} = \cup_{i<m} \mathbf{V}_i$. Then $(\mathbf{H}, \mathbf{T})$ is also a head-tail vertex cover and the third inequality follows from Theorem 2, too. $\qquad\square$

### B.5 PROOF OF THEOREM 4

*Proof.* Notice that Theorem 2 holds for all directed graphs. This theorem then follows by treating the skeleton and the moral graph as directed graphs with loops, i.e., an undirected edge $X_i - X_j$ is treated as two directed edges $X_i \to X_j$ and $X_j \to X_i$. $\qquad\square$

## C IMPLEMENTATION DETAILS

In this section, we present an algorithm to generate a random DAG with a given rank, a low rank version of NOTEARS and GraN-DAG, and also a description of our experimental settings.

### C.1 GENERATING RANDOM DAGS

In Section 4.3, we briefly discuss the idea of generating a random DAG with a given rank. We now describe the detailed procedure in Algorithm 1. In particular, we aim to generate a random DAG with $d$ nodes, average degree $k$, and rank $r$. The first part of Algorithm 1 after initialization is to sample a number $N$, representing the total number of edges, from a binomial distribution $\mathcal{B}(d(d-1)/2, p)$

---

**Algorithm 1** Generating random DAGs

---

**Require:** Number of nodes $d$, average degree $k$, and rank $r$.
**Ensure:** A randomly sampled DAG with the number of nodes $d$, average degree $k$, and rank $r$.
1: Set $M =$ empty graph, $M_p = \varnothing$, and $R = \{(i, j); \ i < j, \ i, j = 1, 2, ..., d\}$.
2: Set $p = k/(d - 1)$.
3: Sample a number $N \sim \mathcal{B}(d(d-1)/2, p)$, where $\mathcal{B}(n, p)$ is a binomial distribution with parameters $n$ and $p$.
4: **if** $N < r$ **then**
5:     **return** FAIL
6: **end if**
7: Sample $r$ indices from $1, \ldots, d - 1$ and store them in $M_p$ in descending order.
8: **for** each $i$ in $M_p$ **do**
9:     Sample an index $j$ from $i + 1$ to $d$.
10:    Add edge $(i, j)$ to $M$ and remove $(i, j)$ from $R$.
11: **end for**
12: **while** $R \neq \varnothing$ and $|M| < N$ **do**
13:    Sample an edge $(i, j)$ from $R$ and remove it from $R$.
14:    **if** adding $(i, j)$ to $M$ does not change the size of the minimum head-tail vertex cover of $M$ **then**
15:       Add $(i, j)$ to $M$.
16:    **end if**
17: **end while**
18: **if** $|M| < N$ **then**
19:    **return** FAIL
20: **end if**
21: **return** $M$

---

where $p = k/(d - 1)$. If $N < r$, Algorithm 1 would return FAIL since a graph with $N < r$ edges could never have rank $r$. Otherwise, Algorithm 1 samples an initial graph with $r$ edges and rank $r$, by choosing $r$ edges such that no two of them share the same head points or the same tail points, i.e., each row and each column of the corresponding adjacency matrix have at most one non-zero entry. Then, Algorithm 1 sequentially samples an edge from $R$ containing all possible edges and checks whether adding this edge to the graph changes the size of the minimum head-tail vertex cover. If not, the edge will be added to the graph; otherwise, it will be removed from $R$. This is because if a graph $\mathcal{G}$ is a super-graph of another graph $\mathcal{H}$, then the size of the minimum head-tail cover of $\mathcal{G}$ is no less than that of $\mathcal{H}$. We repeat the above sampling procedure until there is no edge in $R$ or the number of edges in the resulting graph reaches $N$. If the latter happens, the algorithm will return the generated graph; otherwise, it will return FAIL.

The theoretic basis of Algorithm 1 is Theorem 2. Note that the algorithm may not return a valid graph if the desired number $N$ of edges cannot be reached. This could happen if the input rank is too low while the input average degree is too high. With our experiment settings, we find it rare for Algorithm 1 to fail to return a desired graph.

## C.2 OPTIMIZATION

For this part, we consider a dataset consisting of $n$ i.i.d. observations from $P(X)$ and consequently the expectations in Problems (1) and (2) are replaced by empirical means. Denote the design matrix by $\mathbf{X} \in \mathbb{R}^{n \times d}$, where each row of $\mathbf{X}$ corresponds to an observation and each column represents a variable. Here we use NOTEARS (Zheng et al., 2018) and Gran-DAG (Lachapelle et al., 2020) from each class of methods as examples and will describe their low rank versions in the following. Other gradient-based methods and their optimization procedures can be similarly modified to incorporate the low rank assumption.

---

**Algorithm 2** Optimization procedure for NOTEARS-low-rank

---

**Require:** Design matrix $\mathbf{X}$, starting point $(U_0, V_0, \alpha_0)$, rate $c \in (0, 1)$, tolerance $\epsilon > 0$, and threshold $w > 0$.
**Ensure:** Locally optimal parameter $W^*$.
1: **for** $t = 1, 2, \ldots$ **do**
2:    (Solve primal) $U_{t+1}, V_{t+1} \leftarrow \arg \min_{U,V} L_\rho(U, V, \alpha_t)$ with $\rho$ such that $g(U_{t+1}V_{t+1}^T) < cg(U_t V_t^T)$.
3:    (Dual ascent) $\alpha_{t+1} \leftarrow \alpha_t + \rho g(U_{t+1}V_{t+1}^T)$.
4:    **if** $g(U_{t+1}V_{t+1}^T) < \epsilon$ **then**
5:       Set $U^* = U_{t+1}$ and $V^* = V_{t+1}$.
6:       **break**
7:    **end if**
8: **end for**
9: (Thresholding) Set $W^* = U^* V^{*T} \circ \mathbf{1}(|U^* V^{*T}| > w)$.
10: **return** $W^*$

---

### C.2.1 NOTEARS WITH LOW RANK ASSUMPTION

Following Section 3, the optimization problem in our work can be written as

$$\min_W \frac{1}{2n} \left\| \mathbf{X} - \mathbf{X}UV^T \right\|_F^2, \text{ subject to trace}\left(e^{UV^T \circ UV^T}\right) - d = 0, \tag{5}$$

where $U, V \in \mathbb{R}^{d \times \hat{r}}$ and $\circ$ is the point-wise product. The constraint in Problem (5) holds if and only if $UV^T$ is a weighted adjacency matrix of a DAG. This problem can then be solved by standard numeric optimization methods such as the augmented Lagrangian method (Bertsekas, 1999). In particular, the augmented Lagrangian is given by

$$L_\rho(U, V, \alpha) = \frac{1}{2n} \left\| \mathbf{X} - \mathbf{X}UV^T \right\|_F^2 + \alpha g(UV^T) + \frac{\rho}{2} |g(UV^T)|^2,$$

where $g(UV^T) := \text{trace}\left(e^{UV^T \circ UV^T}\right) - d$, $\alpha$ is the Lagrange multiplier, and $\rho > 0$ is the penalty parameter. The optimization procedure is summarized in Algorithm 2, similar to Zheng et al. (2018, Algorithm 1). Notice that here we do not include the $\ell_1$ penalty term (except for the first and last experiments in Sections 5.1 and 5.5, respectively), for the following reasons: (1) the thresholding procedure can also control false discoveries; (2) we consider relatively sufficient data for the experiments and NOTEARS with thresholding has been shown in Zheng et al. (2018) to perform consistently well even when the graph is sparse; (3) we are more concerned with relatively large and dense graphs, so a sparsity assumption may be harmful, as shown also by Zheng et al. (2018); (4) the $\ell_1$ penalty term requires a tuning parameter, which itself is not easy to choose.

Zheng et al. (2018) used L-BFGS to solve the unconstrained subproblem in Step 2. We alternatively use the Newton conjugate gradient method that is written in C. Empirically, these two optimizers behave similarly in terms of the estimate performance, while the latter can run much faster thanks to its C implementation. The DAG constraint may not be satisfied exactly using iterative numeric methods, so it is a common practice to pick a small tolerance, followed by a thresholding procedure on the estimated entries to obtain exact DAGs. In our implementation, we choose $U_0$ and $V_0$ to be the first $\hat{r}$ columns of the $d \times d$ identity matrices. Other parameter choices are: $\alpha_0 = 0$, $c = 0.25$, $\epsilon = 10^{-6}$, and $w = 0.3$, similar to those used in related methods on the same datasets (e.g., Zheng et al. (2018); Yu et al. (2019); Zhu et al. (2020)). The chosen threshold $w = 0.3$ works well in our experiments and in the experiments of related works that use the same data model. In case the thresholded matrix is not a DAG, one may further increase the threshold until the resulting matrix corresponds to a DAG.

After obtaining $W^*$, we add an additional pruning step: we use linear regression to refit the dataset based on the structure indicated by $W^*$ and then apply another thresholding (with $w = 0.3$) to the refitted weighted adjacency matrix. Both the Newton conjugate gradient optimizer and the pruning technique are also applied to NOTEARS, which not only accelerate the optimization but also improve its performance by obtaining a much lower SHD, particularly for large and dense graphs. See Appendix D.3 for an empirical comparison.

### C.2.2 GraN-DAG with low rank assumption

We next consider a low rank version of GraN-DAG. The optimization problem can be written as

$$\min_{\theta} \quad -\frac{1}{n}\sum_{l=1}^{n}\sum_{i=1}^{d}\log p\left(X_i^{(l)} \mid \mathrm{pa}(X_i, W(\theta))^{(l)}; \theta\right) + \lambda\|W(\theta)\|_* \tag{6}$$
$$\text{subject to } \mathrm{trace}\left(e^{W(\theta)}\right) - d = 0,$$

where $X_i^{(l)}$ is the $l$-th sample of variable $X_i$ and $\mathrm{pa}(X_i, W(\theta))^{(l)}$ means the $l$-th sample of $X_i$'s parents indicated by the adjacency matrix $W(\theta)$. Here, $\theta$ denotes the parameters of neural networks and $W(\theta)$ with non-negative entries is obtained from the neural network path products.

Problem (6) can be solved similarly using augmented Lagrangian. The procedure is similar to Algorithm 2 and is the same to that used by GraN-DAG, with slight modifications: (1) the subproblem in Step 2 is approximately solved using first-order methods; (2) the thresholding at Step 9 is replaced by a variable selection method proposed by Bühlmann et al. (2014). The same variable selection or pruning method is adopted by two other benchmark methods CAM and NOTEARS-MLP in our experiment. Please refer to Lachapelle et al. (2020) and Bühlmann et al. (2014) for further details.

### C.3 Experiment Setup

In our experiments, we consider three data models: linear Gaussian SEMs, linear non-Gaussian SEMs (linear exponential SEMs), and non-linear SEMs (Gaussian processes). Given a randomly generated DAG $\mathcal{G}$, the associated SEM is generated as follows:

**Linear Gaussian**   A linear Gaussian SEM is given by

$$X_i = \sum_{X_j \in \mathrm{pa}(X_i, \mathcal{G})} W(j, i)X_j + \epsilon_i, \quad i = 1, 2, \ldots, d, \tag{7}$$

where $\mathrm{pa}(X_i, \mathcal{G})$ denotes $X_i$'s parents in $\mathcal{G}$ and $\epsilon_i$'s are jointly independent standard Gaussian noises. In our experiments, the weights $W(i, j)$'s are uniformly sampled from $[-2, -0.5] \cup [0.5, 2]$.

**Linear Exponential**   A linear exponential SEM is also generated according to Equation (7), where $\epsilon_i$'s are replaced by jointly independent $\mathrm{Exp}(1)$ random variables. The weights $W(i, j)$'s are sampled from $[-2, -0.5] \cup [0.5, 2]$ uniformly, too.

**Gaussian Processes**   We consider the following additive noise model:

$$X_i = f_i(\mathrm{pa}(X_i, \mathcal{G})) + \epsilon_i, \quad i = 1, 2, \ldots, d, \tag{8}$$

where $\epsilon_i$'s are jointly independent standard Gaussian noises and $f_i$'s are functions sampled from Gaussian processes with RBF kernel of bandwidth one.

We sample $3,000$ observations according the SEM. The reported results of each setting are summarized over 10 repetitions with different seeds. The experiments are run on a Linux workstation with 16-core Intel Xeon 3.20GHz CPU and 128GB RAM.

### C.4 Benchmark Methods

Existing causal structure learning methods used in our experiments all have available implementations, as listed below:

- GES and PC: an implementation of both methods is available through the `py-causal` package at `https://github.com/bd2kccd/py-causal`. We note that, the implementation of `py-causal` package is based on the `CMU TETRAD` project, in which the version of GES is indeed the fast GES algorithm proposed by Ramsey et al. (2017).
- MMHC (Tsamardinos et al., 2006): an implementation is available in the `bnlearn` package at `https://CRAN.R-project.org/package=bnlearn`.

- CAM (Peters et al., 2014): its codes are available through the CRAN R package repository at `https://cran.r-project.org/web/packages/CAM`.

- NOTEARS (Zheng et al., 2018) and NOTEARS-MLP (Zheng et al., 2020): codes are available at the first author's github repository `https://github.com/xunzheng/notears`.

- GraN-DAG (Lachapelle et al., 2020): an implementation is available at the first author's github repository `https://github.com/kurowasan/GraN-DAG`. Note that for graphs of 50 nodes or more, GraN-DAG performs a preliminary neighborhood selection step to avoid overfitting.

- DAG-GNN (Yu et al., 2019): the codes are available at the first author's github repository `https://github.com/fishmoon1234/DAG-GNN`.

- ICA-LiNGAM (Shimizu et al., 2006): an implementation is available at `https://sites.google.com/site/sshimizu06/lingam`.

In the experiments, we mostly use default hyperparameters unless otherwise stated.

## D    ADDITIONAL EXPERIMENTAL RESULTS

### D.1    LINEAR SEMS WITH HIGHER RANKS

This experiment considers graphs of higher ranks. We use rank-specified random graphs with $d = 100$ nodes and rank $r \in \{30, 35, 40, 45, 50\}$ on linear Gaussian SEMs. The results are shown in Figures 10a and 10b with degrees 2 and 8, respectively. We observe that when the rank of the underlying graph becomes higher, the advantage of NOTEARS-low-rank over NOTEARS decreases. Nonetheless, NOTEARS-low-rank with rank $r = 50$ is still comparable to NOTEARS, and has a lower average SHD after removing outlier SHDs using the interquartile range rule.

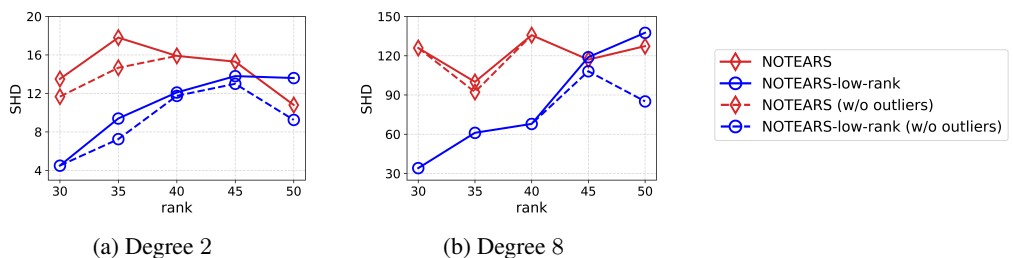

(a) Degree 2                     (b) Degree 8

Figure 10: Average SHDs on rank-specified graphs with higher ranks. The true rank is assumed to be known.

### D.2    NOTEARS-LOW-RANK WITH DIFFERENT SAMPLE SIZES

We next empirically study the consistency of NOTEARS-low-rank. Again, we use rank-specified random graphs (sampled according to Algorithm 1) with $d = 100$ nodes, degree $k = 8$, rank $r = 10$, and linear Gaussian SEMs. We also assume that the true rank is known. We fix the rank parameter $\hat{r} = 10$ and use different sample sizes ranging from 200 to 5,000. From Figure 11, NOTEARS-low-rank performs reasonably well when the sample size is small and tends to have a better performance with a larger number of samples.

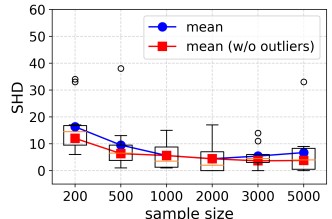

Figure 11: Different sample sizes.

### D.3 FURTHER PRUNING

We compare the empirical results before and after applying the additional pruning technique described in Appendix C.2. The graphs are rank-specified with $d \in \{100, 300\}$ nodes, rank $r = \lceil 0.1d \rceil$, and degree $k \in \{2, 4, 6, 8\}$. We again use linear Gaussian data model with equal noise variances to generate the datasets. The average SHDs are reported in Figure 12. We see that applying an additional pruning step indeed improves the final performance of both NOTEARS and NOTEARS-low-rank, especially on relatively large and dense graphs.

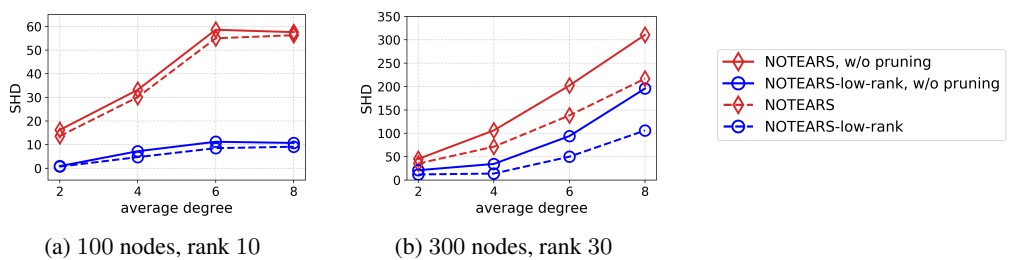

(a) 100 nodes, rank 10          (b) 300 nodes, rank 30

Figure 12: Average SHDs before and after pruning.

### D.4 AN EMPIRICAL COMPARISON BETWEEN ICA-LiNGAM AND DirectLiNGAM

To our best knowledge, there are two Python implementations of ICA-LiNGAM (Shimizu et al., 2006) released by the authors, available at `https://sites.google.com/site/sshimizu06/lingam` and `https://github.com/cdt15/lingam`, respectively, where the latter is a Python package containing several LiNGAM related methods. In the following, we use ICA-LiNGAM-pre and ICA-LiNGAM-cdt to denote these two implementations, respectively. For DirectLiNGAM (Shimizu et al., 2011), we only find a Python implementation available at the previously mentioned Python package containing ICA-LiNGAM-cdt.

Here we run DirectLiNGAM, ICA-LiNGAM-cdt, and ICA-LiNGAM-pre on linear exponential data models with 100-node and rank-10 graphs. The mean SHDs are reported below in Table 1. Based on this experimental result as well as our past experience, DirectLiNGAM usually has a (slightly) better performance than ICA-LiNGAM-cdt, while ICA-LiNGAM-pre has a noticeably (if not much) better performance for relatively dense and large graphs. We are more concerned with relatively large and dense graphs and hence report the results achieved by ICA-LiNGAM-pre in the main paper.

Table 1: An empirical comparison between ICA-LiNGAM and DirectLiNGAM.

| Degree | 2 | 4 | 6 | 8 |
|---|---|---|---|---|
| DirectLiNGAM | 1.4 | 30.1 | 114.2 | 225.0 |
| ICA-LiNGAM-cdt | 2.2 | 37.1 | 128.0 | 241.0 |
| ICA-LiNGAM-pre | 7.0 | 31.7 | 61.7 | 72.6 |

### D.5 DETAILED EMPIRICAL RESULTS FOR EXPERIMENT 1 WITH LINEAR GAUSSIAN SEMS

Table 2 reports detailed results including true positive rates (TPRs), false discovery rates (FDRs), structural Hamming distances (SHDs), and running time on rank-specified graphs with linear Gaussian data model. Here the true rank is assumed to be known and is used as the rank parameter in NOTEARS-low-rank. We also test (fast) GES, MMHC, and PC. However, PC is too slow since some nodes may have a high in-degree (i.e., hubs) in large, dense, and low rank graphs. For the same reason, the skeleton may not be correctly estimated by MMHC, which has a similar performance to that of GES. Therefore, we only include the results of GES for comparison. We treat GES favorably by regarding undirected edges as true positives if the true graph has a directed edge in place of the undirected ones.

Table 2: Detailed results for linear Gaussian data model with equal noise variances.

| **100 nodes, rank 10** | | | | | |
|---|---|---|---|---|---|
| | Degree | 2 | 4 | 6 | 8 |
| NOTEARS-low-rank | TPR | $0.99 \pm 0.02$ | $0.99 \pm 0.03$ | $0.99 \pm 0.01$ | $0.99 \pm 0.01$ |
| | FDR | $0.001 \pm 0.003$ | $0.01 \pm 0.02$ | $0.02 \pm 0.04$ | $0.02 \pm 0.02$ |
| | SHD | $0.7 \pm 1.6$ | $4.7 \pm 10.2$ | $8.5 \pm 13.7$ | $9.1 \pm 11.5$ |
| | Time (mins.) | $1.9 \pm 0.2$ | $5.7 \pm 2.7$ | $5.8 \pm 2.6$ | $7.5 \pm 2.7$ |
| NOTEARS | TPR | $0.90 \pm 0.06$ | $0.90 \pm 0.04$ | $0.87 \pm 0.05$ | $0.89 \pm 0.03$ |
| | FDR | $0.07 \pm 0.03$ | $0.06 \pm 0.04$ | $0.06 \pm 0.02$ | $0.04 \pm 0.02$ |
| | SHD | $13.6 \pm 8.6$ | $30.1 \pm 13.2$ | $55.0 \pm 20.9$ | $56.3 \pm 15.2$ |
| | Time (mins.) | $3.6 \pm 1.6$ | $6.6 \pm 1.6$ | $9.6 \pm 1.5$ | $7.3 \pm 1.5$ |
| fast GES | TPR | $0.3 \pm 0.04$ | $0.13 \pm 0.02$ | $0.08 \pm 0.01$ | $0.06 \pm 0.00$ |
| | FDR | $0.71 \pm 0.03$ | $0.82 \pm 0.03$ | $0.85 \pm 0.01$ | $0.87 \pm 0.01$ |
| | SHD | $141.6 \pm 19.3$ | $292.1 \pm 26.0$ | $412.5 \pm 26.25$ | $521.9 \pm 22.3$ |
| | Time | | < 10 seconds | | |

| **300 nodes, rank 30** | | | | | |
|---|---|---|---|---|---|
| | Degree | 2 | 4 | 6 | 8 |
| NOTEARS-low-rank | TPR | $0.99 \pm 0.01$ | $0.99 \pm 0.01$ | $0.99 \pm 0.01$ | $0.97 \pm 0.05$ |
| | FDR | $0.03 \pm 0.03$ | $0.02 \pm 0.02$ | $0.04 \pm 0.02$ | $0.06 \pm 0.08$ |
| | SHD | $12.0 \pm 10.6$ | $13.9 \pm 14.8$ | $50.0 \pm 32.1$ | $106.0 \pm 169.5$ |
| | Time (mins.) | $57.6 \pm 43.2$ | $76.5 \pm 27.8$ | $158.7 \pm 94.7$ | $262.9 \pm 144.4$ |
| NOTEARS | TPR | $0.94 \pm 0.01$ | $0.93 \pm 0.02$ | $0.93 \pm 0.02$ | $0.91 \pm 0.02$ |
| | FDR | $0.07 \pm 0.02$ | $0.06 \pm 0.03$ | $0.08 \pm 0.04$ | $0.09 \pm 0.06$ |
| | SHD | $35.4 \pm 10.5$ | $71.3 \pm 29.9$ | $138.3 \pm 64.2$ | $216.9 \pm 102.6$ |
| | Time (mins.) | $23.9 \pm 5.8$ | $42.0 \pm 10.5$ | $74.2 \pm 37.9$ | $104.2 \pm 21.7$ |
| fast GES | TPR | $0.31 \pm 0.04$ | $0.12 \pm 0.01$ | $0.07 \pm 0.01$ | $0.05 \pm 0.01$ |
| | FDR | $0.68 \pm 0.05$ | $0.83 \pm 0.02$ | $0.87 \pm 0.01$ | $0.89 \pm 0.01$ |
| | SHD | $427.0 \pm 58.1$ | $883.3 \pm 60.5$ | $1260.8 \pm 59.8$ | $1608.4 \pm 67.5$ |
| | Time | | < 30 seconds | | |

## D.6 DETAILED RESULTS FOR EXPERIMENT 4 WITH NON-LINEAR SEMS

Table 3 reports the detailed SHDs for each method in Section 5.4. We also mark in bold the best results from methods with or without low rank modifications.

Table 3: Detailed SHDs for Experiment 4 with non-linear SEMs.

| Degree | 2 | 4 | 6 | 8 |
|---|---|---|---|---|
| DAG-GNN | $50.1 \pm 8.2$ | $96.8 \pm 11.9$ | $146.9 \pm 10.6$ | $182.0 \pm 13.5$ |
| CAM | $\mathbf{12.9 \pm 3.9}$ | $51.4 \pm 19.3$ | $98.9 \pm 21.7$ | $131.9 \pm 27.1$ |
| NOTEARS-MLP | $19.8 \pm 6.4$ | $\mathbf{47.8 \pm 20.4}$ | $86.0 \pm 15.8$ | $119.4 \pm 23.9$ |
| GraN-DAG | $17.9 \pm 17.0$ | $50.3 \pm 51.7$ | $\mathbf{82.6 \pm 75.8}$ | $\mathbf{109.4 \pm 102.4}$ |
| GraN-DAG, low rank, 0.3 | $20.9 \pm 23.2$ | $45.8 \pm 47.7$ | $65.9 \pm 59.1$ | $87.1 \pm 79.9$ |
| GraN-DAG, low rank, 0.5 | $27.7 \pm 40.8$ | $50.0 \pm 53.4$ | $\mathbf{61.8 \pm 66.7}$ | $83.9 \pm 85.2$ |
| GraN-DAG, low rank, 1.0 | $42.7 \pm 58.8$ | $57.9 \pm 67.6$ | $68.7 \pm 76.2$ | $83.2 \pm 76.9$ |
| GraN-DAG, low rank, validation | $\mathbf{16.0 \pm 4.5}$ | $\mathbf{44.4 \pm 21.0}$ | $63.3 \pm 24.7$ | $\mathbf{77.4 \pm 28.8}$ |

