# OpenReview forum: "On Low Rank Directed Acyclic Graphs and Causal Structure Learning"
_ICLR.cc/2021/Conference — Reject_

### Official Review · AnonReviewer1 · 2020-10-27
**On Low Rank Directed Acyclic Graphs and Causal Structure Learning**

**Rating:** 6
**Confidence:** 3

**Review:**

##########################################################################

Summary:

The paper provides a new approch for learning a (possibly densely-connced) low-rank DAG models in the high dimensional settings. In particular, this paper provides how to exploit the property of the low-rank for recovering a underlying causal structure. Futher shown is that under what circumstance the low-rank assumption holds and heuristically confirms that thgrough simulation settings. Lastly, the proposed approach is compared against the state-of-the-art DAG learning algorithms that requirs the assumption of a sparse graph.

##########################################################################

Reasons for score:


Overall, I vote for accepting. This paper is well-written and delivers its main contribution really well. Futhermore it well summarizes the prior works on learning a causal graph. In addition, the main idea of recovering a graph under low-rank is novel.
However, my major concern is about the simulations of the paper although I acknolwedge that most of relevant papers exploit a similar settings. It would be better to emphasize that the proposed algorithm attempts to learn a complete partial DAG, not a DAG. Although some of related paper fasely asserts that their approches recovers a DAG using conditional independence relationships or score function, I hope this paper clarifies this point.


##########################################################################Pros:
Pros:

1. The paper solve a very important problem of causal inference. It seems to be practical and novel.

2. The paper is really clear and convincing.

##########################################################################

Cons:
One important comment from my side is that the way in which you simulate your models is severely biased. What I always do when I simulate models (and I think others should do something similar), is rescale edge weights for each
node such that if all parents have values with a standard-normal distribution, then the value of the node itself will also have a standard-normal distribution (assuming Gaussian additive noise). In this way one avoids that the variance of the variables blows up (or converges to 0) as one adds more and more nodes to the graph. Therefore, assuming a standard-normal error distribution (or error variance is large) is impractical.

Furthermore, in the densely-connected graph settings, it must be really careful to determine the range of edge weights; otherwise, the variance of the variables are again blowing up. Hence, in some points, the targeted graph is unrealistic in large-scale settings (d is large). Nevertheless, as an emerging field of learning DAG models in polynomial time with complete search, it should be accepted. However, for a better representation and fair comparison, it would be better to change simulation settings.

Lastly, this paper does not explain a complete partial DAG that the proposed method is actually finds. In principle, there might be plenty of solutions for the considered optimization problem. Hence this paper would be clearer for new researcher in DAG model learning if it emphasizes CPDAG or PDAG.

##### Update ######
Although it is responded that the simulation setting used in the paper does cause blowing up samples or marginal variance, it is in general impossible or the setting assumes too sparse case where considered graphs are almost empty. In addition, as you mentioned, I also ackowledge that it is a widely-used setting; however, there are a lot of papers that are rejected because of the unfair simulation setting. I like the main idea of the paper a lot, and hence, I hope the authors set the simulation setting more carefully.

Furthermore, it is really frustaring answer that the authors consider the only case where graph is uniquely idenfiable from the pure observations. As you know that is really rare when the number of nodes is large (p > 50).

---

> ### Author Response · Authors · 2020-11-19
> **Response to Reviewer 1**
>
> We thank the reviewer for a favorable evaluation of our work.
>
> **1. About 'the simulation settings are biased and may cause variance to blow up' and the normalization strategy**
>
> We are grateful for this insightful comment and the suggested normalization strategy. As the reviewer noted, our simulation settings mostly follow related works, for the sake of easy and fair comparisons. In linear and Gaussian Process data models, we find that the dense structures do not make variable values blow up. Yet in our past experience when we considered other data models like time series or polynomial functions, we did encounter the scenarios mentioned by the reviewer. To avoid this, we also applied certain 'normalization' by dividing the variable values by the number of its parents if the variable values blew up. We will definitely try the suggested strategy in our future work.
>
> In addition, we would appreciate pointers to published works using this strategy. We would like to refer to them when we adopt this simulation setting. Thanks.
>
> **2. About 'It would be better to emphasize that the proposed algorithm attempts to learn a complete partial DAG, not a DAG. Although some of related paper falsely asserts that their approaches recovers a DAG using conditional independence relationships or score function, I hope this paper clarifies this point.'**
>
> We believe that this point relates to the identifiability issue in causal structure learning, which depends on the actual SEMs. If the SEM is identifiable and a right score function is used, then the solution to the optimization problem (assuming that we can find the exact optimum) would be consistent, i.e., identical to the true graph with probability $1$ when the sample size goes to infinity. However, if the problem is not identifiable, even given a right score function and that we can find a solution to the optimization problem, we can only hope to find a DAG (a solution must be a DAG due to the acyclicity constraint) that belongs to the Markov equivalence class of the true DAG. In our experiments, we mostly consider identifiable data models so that we can directly compare the estimate with the DAG.
>
> We have added a paragraph in Section 2 to emphasize the issue of identifiability. Thanks for this helpful and insightful suggestion to make our paper more rigorous and readable.

---

> ### Author Response · Authors · 2020-11-25
> **[Response to Reviewer's update] Clarifications about the simulation and experiment setting**
>
> Thanks for the update. We are sorry to see that our earlier responses were regarded as less than adequate. As we said, we are very interested in looking more into the simulation setting recommended by the reviewer, and would appreciate references to that effect. In this paper, the main purpose of the experiments is to demonstrate the gains of taking advantage of a low rank assumption in some state-of-the-art algorithms, so we have generally used the setup and data types on which those algorithms were shown or claimed to work well. We acknowledge the reviewer's point that blow-up of marginal variances may affect performance (e.g. we did observe that the performamce of constraint-based methods such as PC depends on the magnitude of edge weights),but we believe it does not bring any unfair advantage to our low-rank algorithms, and so does not undermine our purpose of demonstrating **comparative** advantages.
>
> In particular, while the variance can be big, the variable values did not blow up. In the linear data models, the weights were uniformly sampled from $[-2, -0.5]\cup[0.5,2]$. For the 100-node graphs with average degree 8, 9 of 10 datasets have absolute values (across all variables and samples) bounded by 250 and the rest dataset has absolute values bounded by 400. Indeed, most variable values (across all samples) are small. In NOTEARS and our method, the goal is to minimize an objective, like MSE, subject to the acyclicity constraint and this scale of variable values did not affect the methods. ICA-LiNGAM relies on the ICA algorithm and the variable values did not cause a problem, either.
>
>
> Regarding the point about identifiability, we hasten to note that for ANM [1], LiNGAM [2], and Linear Gaussian models with equal noise variances [3], the identifiablity is established, **regardless of the number of nodes and the DAG structure** (so the true graph may still have a large Markov equivalence class). We used these provably identifiable models in our experiments to simplify evaluations of performance and facilitate comparisons between different algorithms. We agree with the reviewer that for some of the other commonly used models, such as linear Gaussian models without assuming equal noise variance, we can only hope to discover a CPDAG, and we are now running additional experiments on these models and will use metrics like CPDAG-SHD (by converting the obtained DAG to the corresponding CPDAG as final estimate) to evaluate the performance. But we would like to reiterate our view that our experiments on those provably identifiable models also served our main purpose well.
>
>
> Again, we thank the reviewer for helpful suggestions. We hope what we said above addressed the reviewer's remaining concerns.
>
>
>
> [1] Peters J, Mooij J M, Janzing D, et al. Causal discovery with continuous additive noise models[J]. The Journal of Machine Learning Research, 2014, 15(1): 2009-2053.
>
> [2] Shimizu S, Inazumi T, Sogawa Y, et al. DirectLiNGAM: A direct method for learning a linear non-Gaussian structural equation model[J]. The Journal of Machine Learning Research, 2011, 12: 1225-1248.
>
> [3] Peters J, Bühlmann P. Identifiability of Gaussian structural equation models with equal error variances[J]. Biometrika, 2014, 101(1): 219-228.

---

### Official Review · AnonReviewer3 · 2020-10-28
**An engaging paper in need of improvement**

**Rating:** 5
**Confidence:** 4

**Review:**

# Review
## Summary of the paper and the review
- Paper summary: The paper explores the possibility of exploiting the potential low-rank nature of the underlying causal graph when conducting causal structure learning. Building on the recent gradient-based methods in (causal) DAG selection, they describe ways of integrating this assumption into the algorithms for existing gradient-based methods that assume linear or non-linear structural equation models (SEMs). They also provide some upper and lower bounds to the graph rank given the knowledge of certain structural priors. They test their proposal by augmenting two existing gradient-based methods and comparing its performance to the originals and other methods in simulated data.

- Review summary: The authors examine an intriguing and timely phenomenon, and provide some engaging arguments and results. However the study has some problems in its argumentation and experimentation. I have listed the perceived strengths and weaknesses of the paper below, including my questions for further clarification. I am looking forward to authors' improvements and responses regarding these issues.

## Strengths of the study
- The field of causal discovery can be considered as much a discussion regarding the nature of causality and how it manifests itself in observed joint distributions as it is a discussion regarding specific methods. I believe that it is very important that this work investigates the properties arising from the nature of the underlying distributions and how this should relate to the models and algorithms.
- The authors discuss the structure of causal-DAGs in relation to widely observed complex network structures such as scale-free networks. Although the connections they put forth are not always very clear (see below), I believe that discussing the nature of causal structures in the real world is an important direction, since causal discovery inevitably requires certain structural priors (e.g. independent causal mechanisms).
- Gradient-based DAG discovery methods constitute a promising direction towards causal structure discovery, especially in the presence of a large number of variables. The authors propose a method that can potentially improve any such method with relative ease.
- The modification the authors suggest comes with acceptable computational cost. Considering an extensive potential hyperparameter search, this cost is not negligible, yet it can still be said that it only adds a scaling factor to the complexity.
- Their method can be applied to both linear and nonlinear SEM's. Although the conceptual interpretations in the two case do not always translate, this is more due to the differing nature of the optimization problem in the two cases.
- The authors provide lower and upper bounds for the rank estimations of the causal systems given the existence of certain structural priors.
- The authors conduct simulated experiments where they vary a number of experimental conditions.
- The paper is well-written.

## Weaknesses of the study
### The claimed ubiquity of low-rank structure in causal systems:
- The authors methods aim to exploit underlying low-rankness in the (adjacency matrix of the) underlying causal graphs. When arguing when real life causal systems are likely to be low rank, the authors invoke the prevalence of scale-free networks in real life. This argument is confusing since the observation of scale-free networks are frequently made in relational contexts where a rich-get-richer phenomena are frequently assumed to drive the scale-free property (e.g. a popular social media personality becoming more and more connected as opposed to a typical user).
    - I fail to see why we would also see this in a causal-DAG setting as well, why would we have a causal hub? The authors write that: "It is observed that many real-world networks are scale-free, and some of them may be viewed as causal networks" - the authors could at least present some examples from the literature (in addition to the Pathfinder and arth150 data they cite) as well as more reasoning to make a more persuasive case.
- This might have been less of an issue if the authors showed that low-rank assumption would benefit performance even in the case of high rank causal networks, but their results do not provide strong arguments to that effect (Appendix D.1).
- However, though being a weak point in their argumentation, this does not rule out the potential benefits of their methods, since the fact that the low-rankness assumption does not hold can be established in the cross-validation phase of a study using their method.

### Upper and lower bounds on the rank of the adjacency matrix
- The paper devotes a whole section to extracting lower and upper bounds from prior knowledge regarding the structure of the graph, however provide little as to how we would realistically have access to such information. Almost none of the structural information required by the theorems seem to be easily accessible enough. The authors mention that "we usually have access to some structural information" but do not go into detail.
- Even when we have some structural information, it is not always clear how this information can be used in constraining the search space when this information is anything less than the knowledge of the causal structure itself. For example, even if we knew that our causal network had a number of causal hubs and thus is likely to be scale-free (which we perhaps could in some cases), it still is not clear how this knowledge could be directly used to obtain upper bounds. The relation is even more unclear for the nonlinear case since there is no direct way of relating $\hat{r}$ and $\lambda$.
- This might not have been as big an issue if this was not a section supporting a central claim in an application oriented paper - indeed, the theorems are presented as guidelines to constrain the search space for the estimated rank of the adjacency matrix, so more discussion and demonstration of their practical use is naturally sought.
- I believe that the authors could do two things:
    - They could show how the prior information required in Section 4 can actually plausibly be known beforehand in realistic scenarios, and maybe show example(s) of this in one or more of their applications (unlike in Section 5.3 where the prior information is assumed to be known).
    - Or they could make it clear that these are, though potentially useful as is or after contribution by other research, mostly theoretical demonstrations, and as of yet, their method would likely require extensive hyperparameter search as to the rank of the decomposition or the nuclear norm coefficient. This could possibly require them reevaluate their emphasis on these theoretical findings when introducing their research.

### Experiments
- Given the above concerns, the experiments take on more importance in demonstrating the utility of the methods proposed. Though the experiments have some important findings, a stronger experiments section would allow the authors to make a stronger claim and base their argument more empirically. At least _some_ additional experiments should be feasible given the running times of the algorithms provided in the Appendix.
- Given that it is not clear why causal networks would be both low rank and dense, and given that the prior information to constrain the search space for the rank are not always accessible, I believe that a comparison with sparsity-inducing regularizers such as $\ell_1$ or $\ell_2$-regularizers  would be beneficial, with a correspondingly extensive hyperparameter search so that empirical superiority of the authors' method over others could be empirically established (This has only been conducted in Section 5.5, see below regarding these findings). Given that both would require extensive parameter search, for a practitioner it might not matter whether their regularizer is sparsity or low-rank-decomposition inducing.
- The authors conducted experiments by applying their methods to modify NOTEARS and GraN-DAG. In the absence of application of their modification to other methods, justification for their method choice and why similar improvements should be expected for others as well would be helpful if provided.
- Additional experiments when the true rank goes beyond $d/2$ (for both linear and nonlinear SEMs) would be beneficial in understanding the behavior of the given method in such cases.
    - Also, this would demonstrate a potential contrast between the two methods: in the nonlinear case the cross-validation could produce a very small $\lambda$ to practically disregard the low-rank assumption. However in the linear case, e.g. setting $\hat{r} = r$ would lead to full rank assumption but would lead to $2d^2$ parameters, potentially leading to worse results than the original algorithm both in terms of performance and computation time.
    - In the cases where $d/2 \ll r$, could score comparison e.g. with an unfactorized NOTEARS method lead to the correct choice (that is, full-rank NOTEARS)?
- Is there a specific reason for the absence of results with alternative $k$'s and with ICA-LiNGAM in Linear SEM experiments in scale-free networks in Section 5.2? If not, the inclusion of these would also be illustrative, especially given the latter's comptetive performance in previous experiments.
- Given the centrality of scale-free graphs in their argumentation, is there a specific reason for the lack of nonlinear-SEM experiments with scale-free graphs?
- Section 5.3: As noted before, it might be hard in many cases to provide realistic lower and upper bounds for the rank values. This might require an extensive search for $\hat{r}$ and $\lambda$. Given that the method requires a potentially large parameter search, its effect to general algorithm time complexity must also be mentioned, e.g. its runtime being at least 10x that of the base algorithm. However, I believe that this extra computation cost could be acceptable for increase in structure discovery accuracy.
- The results presented in Figure 7 seem to include very wide interquartile ranges: are the differences between different methods/hyperparameters significant?
    - Also in the description of this experiment, the authors note that the results show the "utility of the low-rank assumption" when the rank is not as low as assumed, but the results show almost equal results, and given that the assumed rank is not much lower than the real rank, I cannot see how this result supports their conclusions, especially given that this is conclusion is presented in the abstract.

## Other comments
- In Figures 2, 3, and 6 the horizontal axis reads "Degree" and the corresponding text mentions "graph degree", in Sections 4.3 and 5.1 . The authors must have meant average (in + out) degree as they did in Section 5.2 or Appendix C.1. I believe the paper would benefit from this being explicit everywhere. In Appendix A the authors seem to equate "graph degree" with total edges in the graph, but they must mean the average number of edges per node in the graph, so that it is consistent with the rest of the paper.
- In Figure 2a, is average NOTEARS (w/o outliers) performance non-existent in the figure or follow exactly the same trajectory as NOTEARS?
- At Pg. 6 in Experiments section the authors might have wanted to cite (Zheng et al. 2018) for the algorithm NOTEARS.
- Figure 5 would benefit from adding the NOTEARS result from the Figure 3.
- Is there a specific reason for authors preferring ICA-LiNGAM over the following DirectLiNGAM?
- Parameter search spaces: In this case $\hat{r}$ would have to be searched for in $[1, d/2]$ (vs. the original algorithm with no low-rank assumption, see above), however it is unclear what the search space for $\lambda$ should be.  Also the authors would benefit from suggesting a search strategy for such a case.
- As a future work, the research could benefit from comparing the inductive bias of the current method to the literature on structure priors [1] or parameter priors affecting model selection through marginal likelihood [2] in DAGs.

## References
1. Eggeling, Ralf, Jussi Viinikka, Aleksis Vuoksenmaa, and Mikko Koivisto. 2019. “On Structure Priors for Learning Bayesian Networks.” In _The 22nd International Conference on Artificial Intelligence and Statistics_, 1687–95. [http://proceedings.mlr.press/v89/eggeling19a.html](http://proceedings.mlr.press/v89/eggeling19a.html).
2. Silander, Tomi, Petri Kontkanen, and Petri Myllymaki. 2007. “On Sensitivity of the MAP Bayesian Network Structure to the Equivalent Sample Size Parameter.” in _Proceedings of the Twenty Third Conference on Uncertainty in Artificial Intelligence_.

---

> ### Author Response · Authors · 2020-11-19
> **Response to Reviewer 3 [Author Response 3/3]**
>
> **3. Experiments (continued)**
>
> *- The lack of non-linear SEM experiments with scale-free graphs*
>
> This is an acute observation. As far as we know, previous good results on scale-free graphs achieved by gradient-based methods set the power parameter $\gamma=1$, a very special case in contrast to typical power parameters. When increasing $\gamma$ to $[2,3]$, the low rank version of GraN-DAG still performed the best, but only slightly better than original GraN-DAG, CAM, and NOTEARS-MLP---the result could not pass a statistical test with significance level $0.1$. We conjectured that this case is currently hard for every extant approach so did not report this result.
>
> *- About the computational costs of searching hyperprameters*
>
> Thanks for this suggestion. In Section 5.3, we have briefly discussed the time complexity, which depends on the number of candidate parameters and the complexity of the base algorithm. We believe that the validation strategy is frequently adopted for tuning hyperparameters, like the $\ell_1$ penalty for NOTEARS-L1.
>
> *- About the wide interquartile range in Figure 7*
>
> We use Student's t-test to test whether the differences are significant. The results show that, with the significance level $\alpha=0.1$,  the results of $\hat{r}=44$ and validation are significantly better than NOTEARS, while the results of $\hat{r}=36$ and $\hat{r}=40$ are as good as NOTEARS. In addition, NOTEARS is significantly better than NOTEARS-L1.
>
> *- Regarding 'the authors note that the results show the "utility of the low-rank assumption" when the rank is not as low as assumed, but the results show almost equal results, and given that the assumed rank is not much lower than the real rank.'*
>
> Here 'rank' in the sentence 'the rank is not as low as assumed' means the true rank of the underlying DAG. We intended to say a low rank adaption, given a good estimate or by validation strategy, can be competitive even when the rank of the underlying causal graph is not very low. In general, for a $100$-node graph, our experiments show that the low rank adaptation works noticeably better when the true rank is greater than $40$. We have clarified the corresponding sentence by replacing 'rank' with 'the rank of the underlying DAG' or 'the true rank'. Thanks for drawing our attention to this need of clarification.
>
> **4. Other comments**
>
> *- About consistent use of 'graph degree'*
>
> Thanks. We have unified the use of graph degree in both the text and figures.
>
> *- 'In Figure 2a, is average NOTEARS (w/o outliers) performance non-existent in the figure or follow exactly the same trajectory as NOTEARS?'*
>
> Yes. No SHDs were considered as outliers by the IQR rule.
>
> *- 'Is there a specific reason for authors preferring ICA-LiNGAM over the following DirectLiNGAM?'*
>
> This is a good question. We are not sure if anyone else has spotted the difference among the implementations of LiNGAM related methods. To our knowledge, there are two Python implementations of ICA-LiNGAM released by the authors, available at https://sites.google.com/site/sshimizu06/lingam and https://github.com/cdt15/lingam, respectively, where the latter is a Python package containing several LiNGAM related methods. In the following we use ICA-LiNGAM-pre and ICA-LiNGAM-cdt to denote these two implementations. For DirectLiNGAM, we only find a Python implementation at the previously mentioned Python package containing ICA-LiNGAM-cdt (remark: the link of DirectLiNGAM at https://sites.google.com/site/sshimizu06/lingam is directed to the Python package). Based on our past experience, DirectLiNGAM usually has a (slightly) better performance than ICA-LiNGAM-cdt, while ICA-LiNGAM-pre has a noticeably (if not much) better result for relatively dense and large graphs. We have rerun all the three algorithms on 100-node graphs with linear exponential data models, and the mean SHDs are given below:
>
> |Average degree|      2      |      4        |      6         |   8       |
> |:---                     |      -:         |   -:    |     -:      |    -: |
> |DirectLiNGAM    |1.4   |  30.1  |  114.2  | 225.0|
> |ICA-LiNGAM-cdt  |  2.2   | 37.1  | 128.0  | 241.0|
> |ICA-LiNGAM-pre    | 7.0   |  31.7   | 61.7    |72.6|
>
> We are more concerned with relatively large and dense graphs and hence report the best results achieved by ICA-LiNGAM-pre. We have added a footnote on Page 7 and Appendix D.4 to make this issue explicit in the paper.
>
> *- About '... it is unclear what the search space for $\lambda$ should be. Also the authors would benefit from suggesting a search strategy for such a case.'*
>
> We believe that the search strategy for $\lambda$ is similar to that for an $\ell_1$ penalty term. In our experiments, we only chose $\{0.1, 0.2, 0.3, 0.5, 1, 2, 5\}$. If time permits, we will try more parameter choices. A practical strategy is to first apply these choices and if a small (or large) choice tends to be better, then we may try several smaller (or larger) values.

---

> ### Author Response · Authors · 2020-11-19
> **Response to Reviewer 3 [Author Response 2/3]**
>
> **3. Experiments**
>
> *- About 'a comparison with sparsity-inducing regularizers such as $l_1$ or $l_2$ -regularizers would be beneficial.'*
>
> This is a very good point, which R4 also made. Originally, we did not include the experiment of NOTEARS with L1 penalty (NOTEARS-L1 in short) for the following reasons (stated in Appendix C.2.1): (1) the thresholding procedure can also control false discoveries;  (2) we consider relatively sufficient data for the experiments and NOTEARS with thresholding has been shown in Zheng et al. (2018) to perform consistently well even when the graph is sparse; (3) we are more concerned with relatively large and dense graphs, so a sparsity assumption may be harmful, as shown also by Zheng et al. (2018); (4) the $\ell_1$ penalty term requires a tuning parameter, which itself is not easy to choose.
>
> Additionally, we have conducted experiments of NOTEARS-L1 on the first experiment with $100$-node graphs and the results have been reported in Figure 3 (a) (the experiments with $300$-node graphs are time-consuming and hopefully we can obtain the results within this rebuttal period). Here we tried different $\ell_1$ penalty weights in $\{0.01, 0.02, 0.05, 0.1, 0.2, 0.5}$, and instead of relying on a validation method, we treated NOTEARS-L1 favorably by picking the lowest SHD with different penalty weights (of course, we cannot adopt this strategy in practice but the result would serve as a lower bound). In most cases (except $2$ out of $10$ cases when the average degree is $2$), the lowest SHD is achieved by NOTEARS-L1 with penalty weight $0.01$. Specifically, the mean SHDs of NOTEARS-L1 are $14.0$, $39.9$, $78.3$, $118.4$ for degree in $\{2, 4, 6, 8\}$, respectively, while the mean SHDs of NOTEARS are $13.6$, $30.1$ $55.0$ and $56.3$. We have added a discussion on NOTEARS-L1 in Section 5.1. Thanks for this comment, which leads to more informative discussions in the revised version of the paper.
>
> *- About 'the authors conducted experiments by applying their methods to modify NOTEARS and GraN-DAG. In the absence of application of their modification to other methods, ... why similar improvements should be expected for others as well would be helpful if provided.'*
>
> In our paper, we picked NOTEARS and GraN-DAG based on their relatively superior performance on respective datasets. Another reason is from the implementation perspective: GraN-DAG is implemented using PyTorch which can readily incorporate the $\ell_1$ penalty, while several other methods use TensorFlow that does not support an $\ell_1$ norm in the objective to be optimized. We believe our adaptations are likely to work well for other methods, as they have a similar form of the optimization problem. Also, the matrix factorization and the nuclear norm regularizer have been shown to be effective in many low rank applications.
>
> *- Additional experiments when the true rank goes beyond $d/2$*
>
> Previously we also conducted experiments with high-rank graphs (up to $\lceil 0.8d \rceil$) on linear data models. The results indicated that the matrix factorization adaptation is less competitive than original NOTEARS in general, even given the true rank. As mentioned by the reviewer, in the matrix factorization method, the number of parameters is $2\hat{r}d$, which would be larger than that of NOTEARS if $\hat{r}$ goes beyond $d/2$. We believe that this is the reason that makes the low rank adaptation less competitive for high-rank graphs. Nevertheless, such a case can be spotted by validation, and we may then use original NOTEARS to handle it. Meanwhile, we empirically find that if a high-rank graph could be transformed into a low-rank one by removing only a few edges, then NOTEARS-low-rank tends to achieve a better performance than original NOTEARS. However, the result was not yet stable so we would like to investigate this issue further in future work.
>
> Following the reviewer's suggestion, we hope to add a discussion on these results along with other experimental results. Unfortunately, the previous experiments with high-rank graphs were not systematically conducted. We may not be able to provide a comprehensive result within this rebuttal period as the experiments of NOTEARS and NOTEARS-low-rank on high-rank graphs are time-consuming, but we will try to add such results later.
>
> *- Alternative k's and with ICA-LiNGAM in linear SEM experiments with scale-free graphs*
>
> We mainly aimed at showing the usefulness of the low rank assumption, so we only considered NOTEARS and a typical value of $k$, while trying to make the experiments informative and representative. We agree that adding these results would enrich the paper. Currently we only have one workstation available, so we opted for the experiments with L1 penalty, which were recommended by several reviewers.

---

> > ### Author Response · Authors · 2020-11-19
> > **References**
> >
> > [5] Xun Zheng, Bryon Aragam, Pradeep Ravikumar, and Eric P. Xing. DAGs with NO TEARS: Continuous optimization for structure learning. In *Advances in Neural Information Processing Systems (NeurIPS)*, 2018.

---

> ### Author Response · Authors · 2020-11-19
> **Response to Reviewer 3 [Author Response 1/3]**
>
> We admire the especially detailed review from the reviewer and appreciate the many insightful and constructive suggestions/comments, which lead to a much improved paper. Below we attempt to address the reviewer's concerns.
>
> **1. On the alleged ubiquity of low-rank structure in causal systems, particularly scale-free ones**
>
> This is a good question. We believe that many empirical studies support this claim. For example, Guelzim et al. (2002) studied the topological and causal structure of the yeast transcriptional regulatory network and showed that this network is scale-free; Gao and Ren (2013) investigated the topology of a causal network for the Chinese financial system and also showed that it is scale-free. Another example is from an on-going project of ours, about finding root causes among alarms where it is usually found that a single root cause has tens or even hundreds of alarms as its causal children.
>
> To explain the ubiquity of scale-free networks and hubs in causal systems, Barabasi and Oltvai (2004) argued that most networks are the results of growth processes and preferential attachments. As an example, Barabasi and Oltvai (2004) explained why protein networks are usually scale-free. (Note that some protein networks are directed and acyclic since many of the reactions are irreversible, which can be viewed as causal networks.) They also mentioned that the growth and preferential attachment in protein networks are probably due to gene duplication. Duplicated genes produce identical proteins that interact with the same protein partners. Therefore, each protein that is in contact with a duplicated protein gains an extra link. Highly connected proteins have a natural advantage: it is not that they are more (or less) likely to be duplicated, but that they are more likely to have a link to a duplicated protein than their weakly connected cousins, and therefore they are more likely to gain new links if a randomly selected protein is duplicated.
>
> We have added the above discussion in Section 4.3 in the revised manuscript to provide more intuitions behind the scale-free graphs. Thanks for this helpful comment.
>
> **2. Regarding the accessibility of the structural priors for estimating upper and lower bounds**
>
> We thank the reviewer for bringing this issue to our attention, together with two useful suggestions. We have revised the corresponding claims regarding the structural priors.
>
> By the claim that 'we usually have access to some structural information', we intended to say that structural information is more accessible than algebraic information. We have revised the statement as 'structural information, such as graph connectivity, distributions of in-degrees and out-degrees, and an estimate of number of hubs, is sometimes more accessible'.
>
> In many applications, such as biological, medical and financial studies, the structural information (e.g., graph has many hubs) is supported by empirical studies and domain knowledge (Barabasi and Oltvai, 2004). We agree with R3 that knowing that a graph has many hubs may not be enough to determine the rank parameter. Nevertheless, given our theoretical characterizations, such qualitative information can on the one hand well indicate whether a low rank assumption is plausible, and on the other hand provides clues about the ballpark of the true rank. Together with validation methods, we may then hope to use low rank adaptations to find a better estimate, as also mentioned by R3.
>
> For practical examples, we notice that causal tiers are one of the structural priors that are commonly seen in multivariate time series analysis (Andrews et al., 2020). With other frequent assumptions such as Markovian assumptions and instantaneous independence, the hierarchical structure of the graph may carry the information needed in Section 4. As another example, consider a causal network of credit default risk contagion. Each node in the graph corresponds to a financial entity such as individuals, companies and banks. In such a network, it may be the case that individuals and small companies are the roots of the network, and the large banks are usually the leafs. Thus, Theorem 3 may be used for estimating the rank of the network. Besides, Theorem 4 links the bound on ranks to the skeleton and moral graph of the underlying causal DAG; the skeleton and moral graph may be well estimated in certain cases like linear Gaussian models. This result could also be useful for developing hybrid algorithms to learn low rank DAGs.

---

> > ### Author Response · Authors · 2020-11-19
> > **References**
> >
> > [1] Bryan Andrews, Peter Spirtes, and Gregory F. Cooper. On the completeness of causal discovery in the presence of latent confounding with tiered background knowledge. In *The 23rd International Conference on Artificial Intelligence and Statistics (AISTATS)*, 2020.
> >
> > [2] Albert-Laszlo Barabasi and Zoltan N Oltvai. Network biology: Understanding the cell’s functional organization. *Nature Reviews Genetics*, 5(2):101–113, 2004.
> >
> > [3] Bo Gao and Ruo-en Ren. The topology of a causal network for the Chinese financial system. *Physica A: Statistical Mechanics and its Applications*, 392(13):2965 – 2976, 2013.
> >
> > [4] Nabil Guelzim, Samuele Bottani, Paul Bourgine, and Francois Kepes. Topological and causal structure of the yeast transcriptional regulatory network. *Nature Genetics*, 31:60–63, 2002.

---

### Official Review · AnonReviewer2 · 2020-10-29

**Rating:** 6
**Confidence:** 3

**Review:**

This paper attempts to exploit the low-rankness of the adjacency matrix of the DAG in Bayesian network structure learning. The overall framework is similar to NOTEARS, except that the adjacency matrix W is decomposed into low rank components W = UV'. To justify the approach, the paper also includes lower and upper bounds of the rank of DAGs, albeit mostly theoretical and not applicable to real experiments.

The paper is very solid in presenting mathematical facts and detailed algorithms. However, my main concern is about the fact that the algorithm requires knowledge (or guess) about the rank. In fact, the experiments in Section 5 already uses the ground truth rank information in NOTEARS-low-rank. Algorithm 1 is a great resource to be shared in the community, however in principal it shouldn't be needed to perform the experiments in Section 5. If one can gain accuracy benefit even without knowing the true rank, paying extra runtime cost is acceptable (Table 1).

There are also many works on combining low-rankness with sparsity, which I suggest the authors to consider as future steps.

Update:
The authors have explained the issue raised in the review. It's not ideal that the algorithm requires the knowledge of rank beforehand, but it's okay if this point is clearly communicated in the paper. I would keep my current score.

---

> ### Author Response · Authors · 2020-11-19
> **Response to Reviewer 2**
>
> We thank the reviewer for the positive feedback on our work as well as a very interesting suggestion for future work.
>
> **1. Regarding the remark: "... the algorithm requires knowledge (or guess) about the rank. In fact, the experiments in Section 5 already uses the ground truth rank information in NOTEARS-low-rank. Algorithm 1 is a great resource to be shared in the community, however in principal it shouldn't be needed to perform the experiments in Section 5. If one can gain accuracy benefit even without knowing the true rank, paying extra runtime cost is acceptable (Table 1).'**
>
> This is an insightful comment. A primary goal of our experiments is to empirically vindicate the usefulness of the low rank assumption, and thus we need a procedure---Algorithm 1---to control the rank of a generated DAG that is used as ground truth. Some of our experiments assumed knowledge of the true rank, serving as an initial sanity check (after all, if the proposed method did not perform well with exact knowledge of the true rank, then we could hardly expect it to be useful in more realistic cases.) However, some experiments only assumed a rough but somewhat accurate estimate of the rank, and other experiments tried a range of possible ranks to probe the utility of the low rank assumption when the rank is unknown. Estimates of the rank could come from domain knowledge, or even from data (e.g., using data to estimate the skeleton or moral graph and then applying Theorem 4), where we can use an additional validation dataset (or by cross validation if the total dataset is not sufficiently large) to determine the final estimate obtained from different parameters. The validation method can be found in our reply to R4's Q4, and also in Section 5.3 in the revised manuscript. Of course, running additional validation increases the total time, but such an approach is frequently used for selecting hyperparameters, such as the weight for an $\ell_1$ penalty in other causal structure learning methods.

---

### Official Review · AnonReviewer4 · 2020-10-30
**Good justification for learning low-rank SEMs**

**Rating:** 5
**Confidence:** 4

**Review:**

# Summary

The paper develops several useful lower and upper bounds on the rank of DAGs — specifically minimum and maximum rank of all weighted matrices that induce the same DAG — in terms of various graphical properties like head-tail vertex cover, number of non-root and non-leaf vertices. The paper also bounds the rank of DAG in terms of the rank of its skeleton and moral graph. The paper proposes learning low-rank linear or non-linear structural equation models (SEMs) by adding simple norm constraints or matrix factorization to existing SEM learning methods. Through experiments on synthetic and real world data the authors demonstrate that when the underlying SEM is low-rank, exploiting this low-rank assumption in the learning process can lead to better performance. The authors also demonstrate that the rank can be estimated using the obtained bounds from a validation set.

# Strengths

1. The main contribution of the paper is a strong justification for learning SEMs under a low-rank assumption by showing that graphs with many hubs are low-rank. Existing theoretical results for learning SEMs show a polynomial dependence of the sample complexity on the maximum degree of the true SEM. Therefore, learning SEMs subject to rank constraints rather than sparsity constraints can be useful for graphs with hubs.
2. The bounds on the rank of DAGs are generally useful beyond learning SEMs.

# Weakness

1. The paper does not propose any novel algorithms for learning low-rank DAGs, other than merely augmenting existing methods with nuclear norm constraint or using matrix factorization.
2. The method for estimating the rank from the validation set is crude and computationally expensive.

# Questions to address in rebuttal

1. In Figure 2, is degree (x-axis) the maximum degree of a node graphs ?
2. Figure 2 shows that the rank increases with the degree and that the rank is always larger than the degree. Therefore, even for graphs with hubs learning SEMs subject to sparsity constraints might still give better results than learning SEMs subject to rank constraints?
3. More details are needed on how the rank is estimated from the validation set with a complete algorithm.

# Post-rebuttal comments
Hello everyone,

I have read the author's response and I am leaning towards rejection. The paper can be divided into two halves. The first half where the authors obtain bounds on ranks of DAGs is the main contribution of the paper and is clearly interesting. The second half of the paper tries to shoehorn these bounds into an algorithm for learning causal DAGs from observational data which is disappointing and is clearly below standard for the following reasons:

1. The bounds depend on the underlying DAG which is unknown and therefore cannot be estimated from samples. Therefore the authors propose using "structural priors" to obtain these bounds. The authors don't mention where they get these structural priors from. Furthermore the bounds are only useful to restrict the hyper-parameter search space in the matrix factorization approach which is applicable to linear SEMs. These bounds can only be used "qualitatively" to guide selection of regularization penalty in the nuclear norm approach which is necessary for non-linear SEM methods.

2. The theoretical results would still be useful if the authors could adequately demonstrate that for certain family of graphs the maximum degree can be high while the rank can be low therefore learning DAGs subject to sparsity constraints (whose sample complexity depend on the maximum degree) can perform worse than learning DAGs with rank constraints. However, this is not clear since in experiments the authors only show the SHD as a function of "average degree" and not "maximum degree". Figure 2 again compares rank against average degree and not maximum degree.

3. The experiments are only performed in the low-dimensional regime at a fixed sample size (3000 samples and 300 nodes).

---

> ### Author Response · Authors · 2020-11-19
> **Response to Reviewer 4**
>
> We greatly appreciate the reviewer's time and effort. Our detailed response follows.
>
> **1. Regarding the comment: "The paper does not propose any novel algorithms for learning low-rank DAGs, other than merely augmenting existing methods with nuclear norm constraint or using matrix factorization."**
>
> The primary goal of the present work is to show that the low rank assumption is useful for causal structure learning. A secondary goal is to demonstrate the relative ease with which one can adapt some of the state-of-the-art algorithms to take advantage of the low rank assumption. For these purposes, we decided to focus on adapting the two well-studied and effective approaches. That said, we agree with the reviewer that it is probably worth pursuing more novel and efficient methods in future work, based on the rich literature on low rank methods.
>
> **2. On the meaning of the x-axis in Figure 2**
>
> The x-axis indicates the average degree. We have changed 'degree' to 'average degree' in the text and also the figures in the revised manuscript.
>
> **3. Regarding the remark: "Figure 2 shows that the rank increases with the degree and that the rank is always larger than the degree. Therefore, even for graphs with hubs learning SEMs subject to sparsity constraints might still give better results than learning SEMs subject to rank constraints?"**
>
> The degree in Figure 2 means the average degree which describes how sparse (or dense) a graph is. We included a discussion in Appendix A on low rank versus sparsity. We hasten to stress that a low rank DAG is not necessarily sparse, or vice versa.
>
> That said, we appreciate the reviewer's good point about the role of sparsity. R3 also mentioned a similar point and R2 pointed out a direction for future work by combining low rank and sparsity assumptions. Originally, we did not include the experiment on NOTEARS with an $\ell_1$ penalty (NOTEARS-L1 in short) for the following reasons (stated in Appendix C.2.1): (1) the thresholding procedure can also control false discoveries;  (2) we consider relatively sufficient data for the experiments and NOTEARS with thresholding has been shown in Zheng et al. (2018) to perform consistently well even when the graph is sparse; (3) we are more concerned with relatively large and dense graphs, so a sparsity assumption may be harmful, as shown also by Zheng et al. (2018); (4) the $\ell_1$ penalty term requires a tuning parameter, which itself is not easy to choose.
>
> In the last experiment in Section 5.5, we included NOTEARS-L1 for a simple comparison. For more details, we have additionally conducted experiments of NOTEARS-L1 on the first experiment with $100$-node graphs and the results have been reported in Figure 3 (a) (the experiments with $300$-node graphs are time-consuming and hopefully we can obtain the results by the end of the rebuttal period). Here we tried different $\ell_1$ penalty weights in $\{0.01, 0.02, 0.05, 0.1, 0.2, 0.5}$, and instead of relying on a validation method, we treated NOTEARS-L1 favorably by picking the lowest SHD with different penalty weights (of course, we cannot adopt this strategy in practice but the results serve to show that NOTEARS-L1 does not improve NOTEARS in our setting). In most cases (except $2$ out of $10$ cases when the average degree is $2$), the lowest SHD is achieved by NOTEARS-L1 with penalty weight $0.01$. Specifically, the mean SHDs of NOTEARS-L1 are $14.0$, $39.9$, $78.3$, $118.4$ for degree in ${2, 4, 6, 8}$, respectively, while the mean SHDs of NOTEARS are $13.6$, $30.1$ $55.0$ and $56.3$. We have added a discussion on NOTEARS-L1 in Section 5.1. Thanks for this suggestion to make our paper more informative.
>
> **4. About the details on how to estimate ranks from the validation set**
>
> Thanks for this suggestion. We have added more details in Section 5.3. Basically, we first split the total dataset into a training and a validation dataset. Then, given a lower bound and an upper bound of the true rank, we select $7$ evenly distributed rank parameters and learn a DAG for each of them based on the training dataset. We use the validation set to evaluate each learned DAG and choose the one with the best score function as our estimate. Here the validation set is not directly for estimating the rank but for selecting the causal graph. We could treat the rank of the selected graph as an approximation to the true rank; however, currently we do not have a rigorous result regarding the discrepancy between this estimate and the true rank.
>
> **References**
>
> [1] Xun Zheng, Bryon Aragam, Pradeep Ravikumar, and Eric P. Xing. DAGs with NO TEARS: Continuous optimization for structure learning. In *Advances in Neural Information Processing Systems (NeurIPS)*, 2018.

---

### Author Response · Authors · 2020-11-19
**General Response**

We appreciate the comments/suggestions from the reviewers that have helped us greatly improve the paper. We have uploaded a revised manuscript, taking into account all the suggestions/comments. Below are some notable changes:

- We changed 'degree' to 'average degree' in the text and figures to make its meaning clear, according to R3's and R4's comments
- We added a paragraph at the end of Section 2 to clarify the identifiability issue (about 'DAG and CPDAG'), following R1's suggestion.
- We provided a more detailed discussion on the low rank structure in causal systems in Section 4.3, in response to R3's comments.
- In Section 5.1, we included the empirical results of NOTEARS with an $\ell_1$ penalty (NOTEARS-L1 in short), along with a discussion on the role of sparsity, based on R3's and R4's comments.
- We provided more details on the validation approach in Section 5.3, according to R2's, R3's and R4's comments.
- We also added some future directions in the concluding remarks, as suggested by R2 and R3.
- Following R3's comment, a brief comparison between ICA-LiNGAM and DirectLiNGAM was added in Appendix D.4, to explain the reason for using ICA-LiNGAM in the experiments.

We also decided to add several experiments to provide more insights into the low rank assumption in causal structure learning. We have not yet finished all the experiments due to limited time, and will add these results and discussions later.

We once again thank all the reviewers for the effort they put into reviewing our submission.

---

### Decision · Program_Chairs · 2021-01-07
**Final Decision**

**Decision:**

Reject

**Comment:**

This paper studies the low-rank properties of DAG models, and illustrates through proof-of-concept how low-rank-ness can be exploited in structure learning of DAGs. After a lengthy discussion amongst the reviewers, it became clear that although there are some interesting ideas here, there is not enough enthusiasm for this work in its current form. The results in Section 4 connecting rank to structural properties are interesting, but the reviewers were concerned by the lack of precise statements connecting these results to known ensembles such as scale-free graphs (even though the authors discuss some heuristic connections). In the end, despite considerable enthusiasm regarding these ideas and the importance of the problem studied, there remained too many concerns that require a major revision before acceptance.